# Factor analysis of ancient population genomic samples

Olivier François [1,3✉] & Flora Jay [2,3✉]

The recent years have seen a growing number of studies investigating evolutionary questions using ancient DNA. To address these questions, one of the most frequently-used method is principal component analysis (PCA). When PCA is applied to temporal samples, the sample dates are, however, ignored during analysis, leading to imperfect representations of samples in PC plots. Here, we present a factor analysis (FA) method in which individual scores are corrected for the effect of allele frequency drift over time. We obtained exact solutions for the estimates of corrected factors, and we provided a fast algorithm for their computation. Using computer simulations and ancient European samples, we compared geometric representations obtained from FA with PCA and with ancestry estimation programs. In admixture analyses, FA estimates agreed with tree-based statistics, and they were more accurate than those obtained from PCA projections and from ancestry estimation programs. A great advantage of FA over existing approaches is to improve descriptive analyses of ancient DNA samples without requiring inclusion of outgroup or present-day samples.

[1] Université Grenoble-Alpes, Centre National de la Recherche Scientifique, Grenoble INP, Laboratoire TIMC-IMAG UMR 5525, 38000 Grenoble, France. [2] Université Paris-Saclay, Centre National de la Recherche Scientifique, Inria, Laboratoire de Recherche en Informatique UMR 8623, Bâtiment 650 Ada Lovelace, 91405 Orsay Cedex, France. [3] These authors contributed equally: Olivier François, Flora Jay. ✉email: olivier.francois@univ-grenoble-alpes.fr; flora.jay@lri.fr

Temporal samples of DNA and ancient DNA have become prominent tools in inferring past events in the history of humans and other organisms[1–5]. In such studies, a central question concerns the inference of ancestral relationships between sampled populations[6]. Evolutionary biologists and population geneticists have devised many methods for addressing this question. To visualize ancestral relationships between sampled populations, some representations are based on principal component analysis (PCA), often by projecting ancient samples on axes built from present-day samples, while other representations use admixture estimation programs and statistical measures of treeness[7–10]. All those methods are strongly related to each other[11,12]. In population genetics, PCA is usually performed by finding the eigenvalues and eigenvectors, or axes, of the covariance matrix of allele frequencies. The first axes indicate the directions which account for most of the genetic variation. Individual genotypes are then projected on the space spanned by those axes, offering a visualization of samples in which relative distances reflect ancestral relationships among samples[12]. When PCA is applied to analyze temporal samples, information on sample dates is, however, omitted in the computation of eigenvalues and eigenvectors[13,14].

Previous studies have reported that differences in sample dates could modify the principal axes of a PCA[15], creating spurious sinusoidal shapes similar to those observed with geographic samples[16]. The combination of both time and spatial heterogeneity in sampling further complicate the interpretation of patterns observed in PC plots. Local dispersal through time causes ancient samples to be shrunk toward the center of the PC plot and not to cluster with their present-day counterparts[17]. Similar distortions linked to gradients and longitudinal data occur not only in population genetics, but in other applications, such as multidimensional scaling and ordination analysis, where they are called *horseshoes* or *arches*[18,19]. Alternative PCA methods that combine ancient and modern individuals by projecting ancient samples on present-day samples suffer from additional statistical issues. PC projections exhibit a *shrinkage* bias toward the center of the principal axes which also complicates the interpretation of the data[20]. Since biases could lead to incorrect estimates of individual ancestries or misinterpreting the data, it is important to correct principal components when temporally distinct samples are used in descriptive analyses. Corrections of sinusoidal patterns arising in principal components have been proposed when distortions are caused by spatial auto-correlation in geographic samples[18,21]. Modified versions of the STRUCTURE algorithm—which may be viewed as a constrained version of PCA—were also developed to integrate corrections based on spatial or temporal diffusion models[7,9,22,23], but less efforts have been devoted to correct PCA itself.

In recent studies of ancient DNA, ancestral relationships between samples are most often inferred by using a combination of methods including PC projections, ancestry estimation programs similar to STRUCTURE and *F*-statistics[2,8,24–27]. In the last approach, *F*-statistics are tree-based estimates of population admixture proportions that rely on shared genetic drift between sets of populations[11,27]. While *F*-statistics do not represent individual samples in a reduced geometric space, those quantities should correlate with distances between population centers in ideal geometric representations. In this study, we provide evidence that representations of ancient DNA samples based on PC projections, STRUCTURE or PCA disagree with each other and with *F*-statistics. We present a factor analysis (FA) that enables geometric representations of ancient samples in better agreement with *F*-statistics, and does not use reference populations such as outgroup or present-day samples. We provide evidence that the estimates of ancestry coefficients derived from those geometric

representations are more accurate than the coefficients obtained from PC projections and from ancestry estimation programs. We illustrate our method with several analyses of ancient DNA samples from prehistoric European populations.

## Results

**Factor analysis**. We developed a FA method for representing ancestral relationships among samples collected at distinct time points in the past. The objective of our approach was to propose a factorial decomposition of a data matrix similar to a PCA, in which individual scores are corrected for the effect of allele frequency drift over time. The data, $\mathbf{Y}$, were recorded in an $n \times p$ matrix of genotypes, where $n$ is the number of individual samples and $p$ is the number of markers measured as single nucleotide polymorphisms (SNPs). More generally, the data matrix could result from any preliminary operations, where, for example, undesired effects of genomic coverage have been removed from the observed genotypes (see "Methods" section). The data were centered so that the mean value for each individual is null. Each ancient sample was associated with a sampling date corresponding to the age of the sample. The dates were transformed to scale with the variance of allele frequencies, so that values closest to zero represented the most ancient ages (see "Methods" section). To incorporate correction for temporal drift, we modeled the data matrix as $\mathbf{Y} = \mathbf{W} + \mathbf{ZB}^T + \epsilon$, the sum of a latent matrix, $\mathbf{W}$, plus correction factors and residual errors. In this model, the residual errors, $\epsilon$, consist of $n \times p$ i.i.d. random variables having a univariate Gaussian distribution, $N(0, \sigma^2)$. The matrix of effect sizes, $\mathbf{B}$, is assumed to have an informative prior distribution, in which all coefficients are i.i.d. random variables with a Gaussian distribution, $N(0, \alpha)$. The factors and their loadings are obtained by singular value decomposition (SVD) of the latent matrix, $\mathbf{W} = \mathbf{UV}^T$, once $\mathbf{W}$ is estimated. The number of factors, $K$, can be set to any number smaller than $n$ and $p$ depending on how drastically one wants to reduce the dimension of the data and approximate the data matrix with the latent factors. This number can be set greater or equal than the number of ancestral groups when this number is known. In this case, the factors contained in the first $K-1$ column vectors, $\mathbf{u}_1, \ldots, \mathbf{u}_{K-1}$, of the matrix $\mathbf{U}$ reflect ancestral relationships among samples in the data[7].

We used correction factors related to the covariance function of the Brownian process (Fig. 1a–c). This notion corresponds to the diffusion approximation of allele frequency drift in a random mating population conditional on the non-fixation of alleles in the population[28–30], and underlies the development of several recent methods of ancestry estimation[8,10,11,23]. According to Cavalli-Sforza and Edwards[30], the Brownian motion may be the simplest model in which the allele frequencies are approximately Gaussian with variance proportional to the time elapsed, while the mean remains constant in the absence of selection. In this model, random drift alone determines the variation in gene frequencies, and population size and structure are taken to be constant. In our approach, the correction factors, $\mathbf{Z}$, were obtained after a spectral decomposition of the Brownian covariance matrix, $\mathbf{C}$, defined as $c_{ij} = \min(t_i, t_j)$, for all pairs of individuals $i$ and $j$ and their corresponding transformed time points $t_i, t_j$. More specifically, we computed $\mathbf{Z} = \mathbf{P}\sqrt{\Lambda}$, where $\mathbf{P}$ is a unitary matrix of eigenvectors of $\mathbf{C}$ and $\Lambda$ is the diagonal matrix containing the eigenvalues of $\mathbf{C}$ (see "Methods" section). According to the Karhunen–Loève theorem[31], the eigenvalues of $\mathbf{C}$ can be approximated as $\lambda_i \approx n/(i - 1/2)^2 \pi^2$, and the correction factors as $Z_{ij} \approx f_i(t_j)\sqrt{\lambda_i/n}$ where $f_i(t) = \sqrt{2}\sin((i - 1/2)\pi t)$, for all $t$ in the interval $[0, 1]$ and $i$ in $1, \ldots, n$. According to these results, the correction factors have sinusoidal shapes, and the covariance model is consistent with the

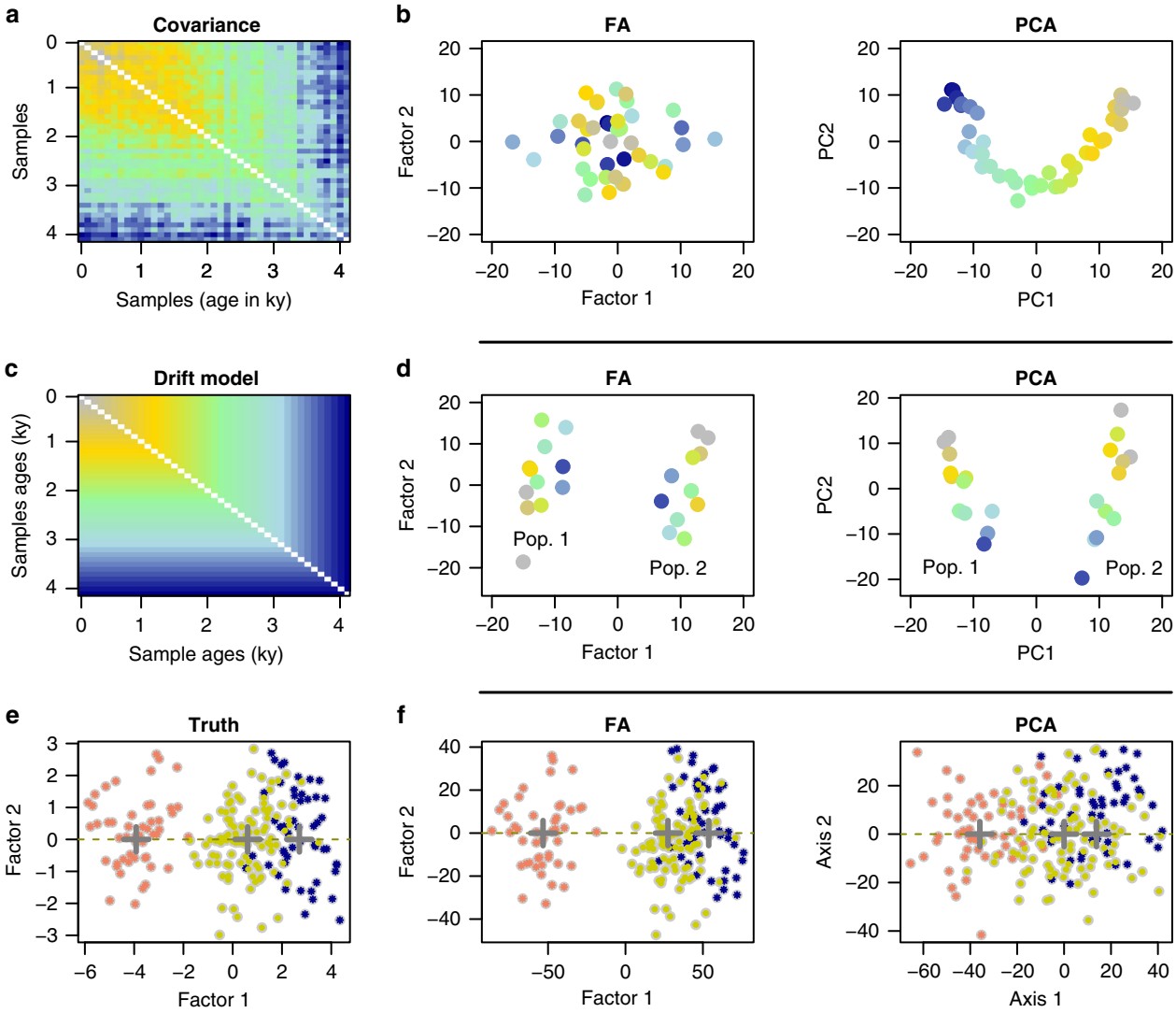

**Fig. 1 Correcting arch and shrinkage effects. a–c** Coalescent simulation of allele frequencies drifting through time in a random mating population. Samples with ages ranging from 0 (present-day) to 4000 years (past) were simulated. **a** Covariance matrix for observed samples. In covariance matrices (**a–c**), blue colors indicate lower covariance values whereas the yellow and gray colors indicate higher values. **b** FA and PC plots of individual samples (present-day: gray color, past: dark blue color). **c** Theoretical covariance matrix defined as the minimum of forward times for pairs of samples (Brownian model). **d** FA and PC plots for a coalescent simulation of a two-population model (left: population 1, right: population 2, present-day: gray color, past: dark blue color). **e–f** Factor analysis corrects for a shrinkage effect visible in projections of ancient samples (salmon and blue colors) onto PCs of present-day admixed individuals (yellow color). Gray crosses represent population centers from which admixture estimates are computed.

arch effect observed in the principal components of the genetic data.

**Factor estimates**. Statistical estimates of the factors, **U**, were obtained by maximizing the posterior distribution in a Bayesian framework. Setting the regularization parameter equal to the noise-to-signal ratio or *drift parameter*, $\lambda = \sigma^2/\alpha$, the factors corresponded to estimates in a latent factor regression model with ridge penalty[32,33]. More precisely, an estimate of the latent matrix **W** was computed as the best approximation of rank $K$ of the matrix **Y** for an appropriate matrix norm. The optimal solution for **W** was computed from an exact formula, and the $K$ factors were obtained from the SVD of **W** (Supplementary Note 1, see "Methods" section). For $p$ much larger than $n$, the complexity of the algorithm is of order $O(npK)$. The running times for FA can be understood from the description of the algorithm in Supplementary Note 1, which requires one application of the SVD to estimate the latent matrix. A second application of the SVD is

then needed to obtain the factors from the latent matrix. In experiments using the SVD algorithm implemented in the *base* library of the R programming language[34], the running times for FA were about twice the running times of a classical PCA.

**Examples of distortions due to genetic drift**. To provide examples of distortion arising in PCA due to uncorrected temporal drift, we performed simulations of a coalescent model for forty-one samples with ages ranging from 0 to 4000 generations in a random mating population with effective size $N_e = 10,000$ (Fig. 1a–b). The patterns observed in the sample covariance matrix were highly similar to those observed in the theoretical covariance model corresponding to a Brownian process (Fig. 1a, c). Both in the Brownian model and in the empirical matrix, pairs of samples that included the most ancient samples had the smallest covariance values. The PC plots of individual samples exhibited sinusoidal patterns, in which the most ancient and recent samples were placed at both extremes of a horseshoe.

Correcting for temporal drift, the FA plot displayed a single cluster grouping all samples without any apparent temporal structure among samples (Fig. 1b), showing that the distortion due to genetic drift was correctly removed in an FA using a Brownian covariance model. In a second series of experiments, we simulated models of divergence between two populations (Fig. 1d). In a PCA of simulated samples, PC1 reflected the level of divergence between populations while PC2 represented temporal drift (Fig. 1d). Correcting for drift, the FA plot exhibited two clusters without any apparent structure within each group (Fig. 1d). In a hundred simulations of this scenario, the Davies-Bouldin clustering index reached higher values in the FA plots than in the PC plots, meaning that the clusters better separated the two populations in FA plots than in PC plots (Supplementary Fig. 1c, "Methods" section). In generative model simulations, factor 1 in FA better explained the hidden factor than did the first PC in PCA (Supplementary Fig. 1d). In a third series of experiments, we considered two-way admixture models in which a population split occurred 1300 generations ago. The two divergent populations came into contact 500 generations ago, giving rise to descendants having 75% ancestry in the first ancestral population and 25% ancestry in the second ancestral population. Using coalescent simulations, one hundred present-day individuals were sampled from the admixed population, and fifty individuals were sampled from each ancestral population before the admixture event (1000 generations ago). Artificial genotypes were also simulated according to the generative model to mimic levels of admixture observed in coalescent models (see "Methods" section). The objective of those experiments was to compare the results of PC projections of ancient samples onto the present-day population with those obtained with FA correcting for drift. Typical plots for PC projections exhibited a shrinkage effect in which the projected samples were shifted toward zero, closer to the admixed population than expected (Fig. 1e, f, Supplementary Fig. 2). The shrinkage effect was more pronounced in coalescent simulations than in generative model simulations (Supplementary Fig. 2). Correction for temporal drift in FA did not exhibit such an undesired effect (Fig. 1f). In the FA plot, the locations of centers of ancestral clusters reflected admixture levels more precisely than in PC plots, and the factor scores had a smaller variance than the PC scores (Supplementary Fig. 2). Both for generative and for coalescent simulations, mean squared errors of admixture proportion estimates computed on the first axis were higher in PC projections than in FA plots Those results showed that correcting for genetic drift in FA improved the representation of admixed individuals and their source populations compared with projections on PCs.

**Comparisons with other methods**. Using two-way admixture models, we compared ancestry estimates obtained from FA with those from PC projections on the present-day samples, with estimates from the program qpAdm[2,8] and with estimates from the STRUCTURE model[9,24], implemented with sparse non-negative matrix factorization[26]. Here qpAdm was considered to be the state-of-the-art method for estimating admixture coefficients from ancient DNA samples[27]. Based on connections between population genetic models and matrix factorization[7,12,25], we performed simulations of ancestral and admixed genotypes using factor models. The generative mechanism was similar to the simulation of multilocus genotypes in STRUCTURE, which is based on a matrix of individual mixture coefficients multiplied by a matrix of allelic frequencies in source populations[9,25]. By replacing allelic frequencies with Gaussian variables and using a probit link function, our generative mechanism had the advantage of providing a ground truth both

for factors and for admixture coefficients, while remaining interpretable in terms of population genetics.

When considering a relatively high level of divergence between ancestral populations, the FA estimates were precise and relatively insensitive to the drift value, similarly to the estimates provided by qpAdm (Fig. 2, $F_{ST} = 25\%$). Both FA and qpAdm estimates were biased in low admixture scenarios, with a lower bias for FA (Fig. 2a). The estimates computed from PC projections and from STRUCTURE exhibited higher statistical errors (Fig. 2d). The lower performances were explained by higher variance in estimates based on PCA (Fig. 2b), and by temporal distortions arising in STRUCTURE estimates (Fig. 2c). Considering a lower level of divergence between ancestral populations did not change the ranking of method performances (Supplementary Fig. 3, $F_{ST} = 5\%$). For weak drift, estimates from PC projections exhibited less bias than the qpAdm estimates while keeping similar levels of statistical error (Supplementary Fig. 3b, bias for PCA: 2.3%, bias for qpAdm: 4.9%, $t = 4.02$, $P = 10^{-4}$). STRUCTURE did not detect admixture in the case of high genetic drift (Supplementary Fig. 3c). We eventually performed simulations for which summary statistics were close to those observed in European Bronze Age populations (Supplementary Fig. 4). In all methods, statistical errors tended to decrease, but estimates from FA and qpAdm were still more accurate than those computed with PC projections and STRUCTURE. Overall the results suggested that the geometric representations of samples provided by FA were accurate and consistent with estimates derived from tree-based approaches. Representations based on PC projections and STRUCTURE were less accurate due to a larger variance and because temporal distortions affect STRUCTURE in a way similar to PCA.

**Analysis of ancient European data**. We used our approach to analyze a merged data set consisting of 143,081 SNP genotypes for 521 present-day European individuals and 704 ancient samples from Eurasia[2–4,35–41]. The samples had ages less than 14,000 years cal BP, and were chosen for their higher level of genomic coverage (Supplementary Fig. 5). Prior to analysis with FA and PCA, genotypes were corrected for the effect of coverage (see "Methods" section). We first computed principal components on the present-day samples, mainly from Great Britain, Italy, Spain, Finland and Russia (Supplementary Data 1 for a list of sample IDs)[40,41]. Following the classical approach, we projected the ancient samples on the first two PCs of the present-day samples (Supplementary Fig. 6). Next, we performed FA with temporal correction for present-day and ancient samples (Supplementary Fig. 7), choosing the drift parameter so that the scores of present-day individuals on factor 1 correlate with their scores on the first principal component (Pearson's squared correlation $\rho^2 = 0.91$, $P < 10^{-10}$, F-test with 1 and 519 df). For this value of $\lambda$, we reanalyzed the ancient samples independently of present-day populations with FA (Fig. 3A). The FA plot revealed a pattern in which most ancient samples were represented as mixtures of three ancestral populations, put at the vertices of a triangle. Factor 1 separated hunter-gatherers from Serbia and Scandinavia from early farmers from Anatolia, while factor 2 corresponded to genetic contributions from the Pontic steppe (Yamnaya culture). In PC projections, the status of the Yamnaya samples as representing one of the three ancestral populations was less clear than in the FA plot (Supplementary Fig. 6). In the FA of all samples, present-day individuals from Great Britain and Finland were closer to the Yamnaya than individuals from Italy and Iberia (Supplementary Fig. 7). For several geographic regions, including Great Britain, Germany, Czech Republic and Hungary,

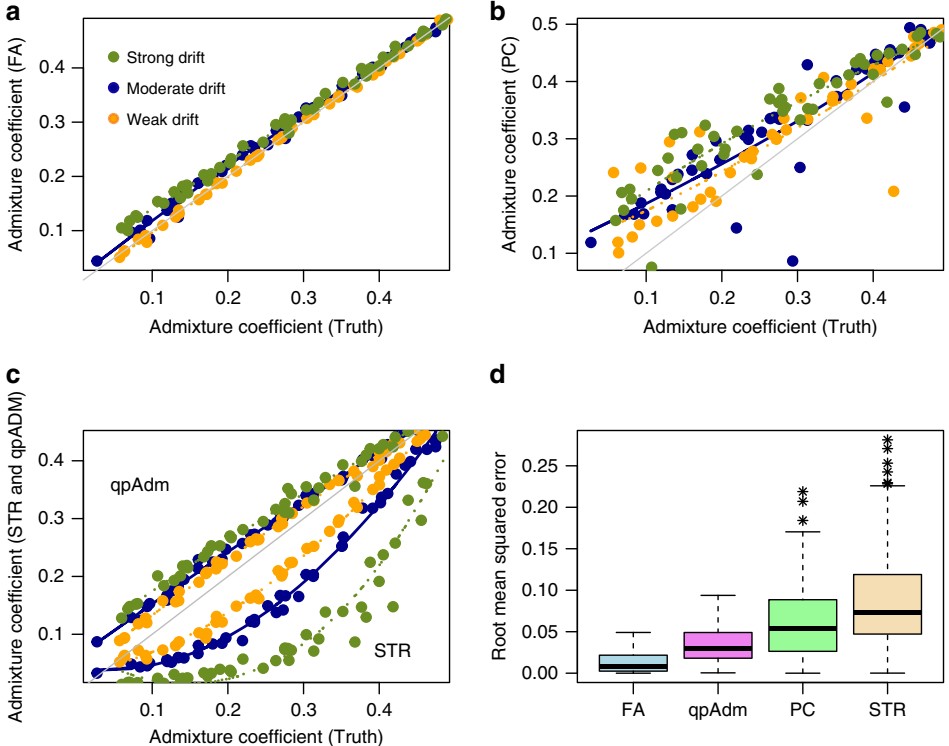

**Fig. 2 Comparisons of ancestry estimates.** Inference of ancestry proportions in two-way admixture models with high level of divergence between the ancestral populations ($F_{ST} = 25\%$). **a** Ancestry estimates obtained from FA for three levels of drift intensity represented in green, blue and orange colors, **b** Ancestry estimates obtained from PC projections on admixed samples, **c** Ancestry estimates from qpAdm (above diagonal) and STRUCTURE (STR, below diagonal), **d** Root mean squared errors for admixture estimates from the four methods. FA: Factor Analysis, PC: Projections on PCs of admixed samples, STR: STRUCTURE estimates computed with non-negative matrix factorization. Boxplots computed on $m = 135$ replicates (minimum, median, interquartile range, 1.5 IQR away from 75th percentile or maximum). Gray line: diagonal, colored line: local regression.

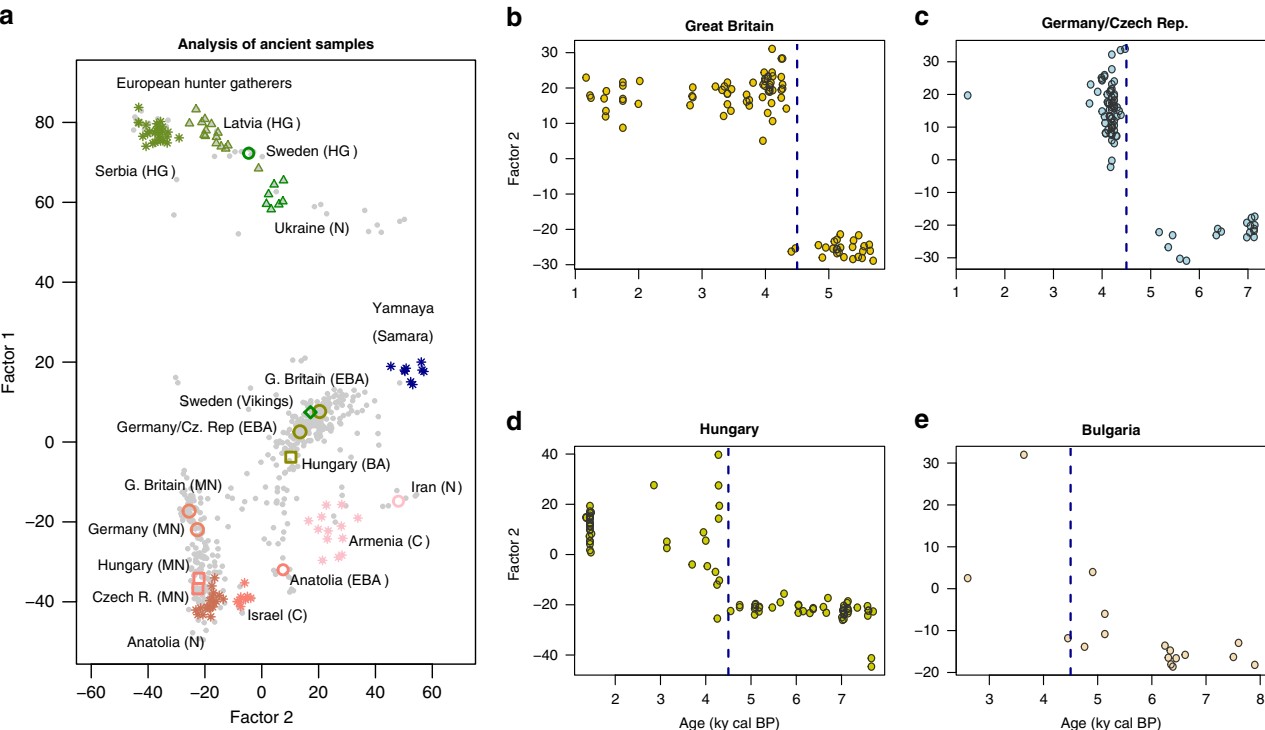

**Fig. 3 Factor analysis of ancient European genomes. a** Factor analysis of 704 ancient Eurasian individuals with ages ranging between 0.4 and 14 ky BP. Factor 1 separates western hunter-gatherers (Serbia) from early farmers (Anatolia), while factor 2 corresponds to steppe ancestry. **b–e** Factor 2 as a function of age for samples from four geographic regions. The FA supports a major change in genetic mixture of samples from Great Britain and Central Europe around 4500 years BP (dashed line). HG: Hunter-Gatherers, N: Neolithic, MN: Middle Neolithic, C: Copper Age, EBA: Early Bronze Age.

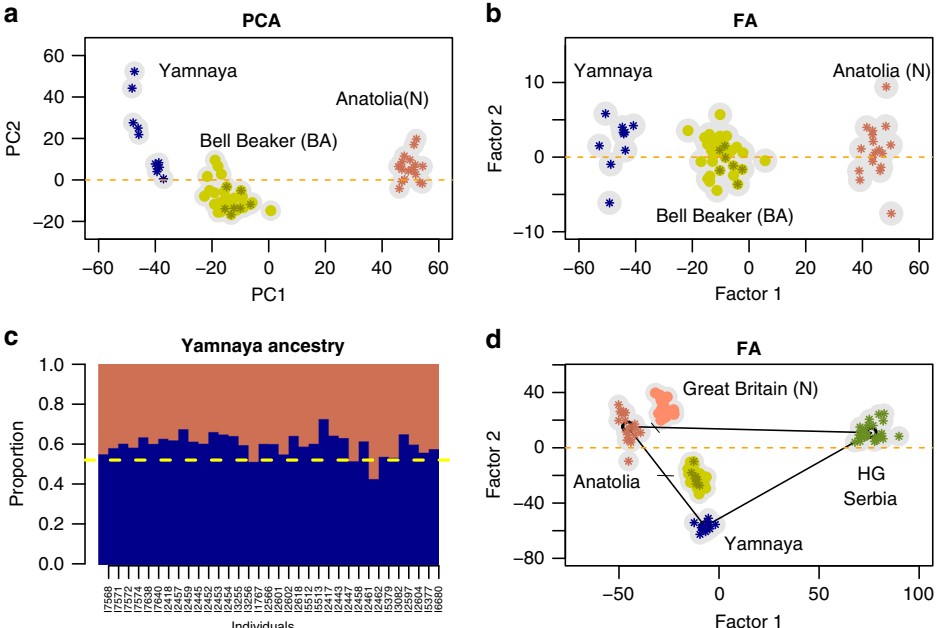

**Fig. 4 Analysis of 34 samples from Bronze Age England. a** The PCA of Bronze Age, Steppe and Early Farmer samples exhibits a horseshoe pattern, **b** FA for the samples, **c** Steppe ancestry estimates for Bronze Age England samples computed from Factor 1 (x-axis: sample IDs). The dashed line corresponds to the estimate computed by the qpAdm program using Yamnaya Samara and Anatolia (N) as ancestors, and Yoruba, Russia Sidelkino (EHG), France-Ranchot (WHG) as outgroups, **d** FA of samples from Bronze Age and Neolithic Great Britain with three ancestral groups. The ticks on the triangle sides indicate the proportions of ancestry from HG and Yamnaya. Samples: EBA (Early Bronze Age): 11, EBA Bell Beaker: 16, MBA (Middle Bronze Age, dark green stars): 7, Great Britain (N) Neolithic England and Scotland: 28, HG (Hunter Gatherers, Serbia): 31.

discontinuities in Yamnaya ancestry were reflected in factor 2 (Fig. 3b–e). These discontinuities supported the hypothesis of a sudden change in the European gene pool, with a strong genetic input from the Pontic steppe around 4500 years ago[2,4]. Genetic discontinuities in Anatolian ancestry were symmetrically reflected in factor 1, which also supported a resurgence of hunter-gatherer ancestry in late neolithic samples from Germany, Czech Republic and Hungary[2] (Supplementary Fig. 8).

**Analysis of samples from Bronze Age England**. To focus on a specific geographic region, we reanalyzed 34 samples from Bronze Age (BA) England and 28 samples from Neolithic (N) England and Scotland, using Anatolia (N), Samara (Yamnaya) and Serbia (HG) as proxies for ancestral populations (Fig. 4). The PCA of BA England, Yamnaya and Anatolian samples exhibited a horseshoe pattern, complicating the evaluation of the relative inputs from Yamnaya and early farmers to BA England samples (Fig. 4a). In an FA correcting for drift, the BA samples aligned with their assumed ancestral groups (Fig. 4b), allowing us to estimate an ancestry coefficient for each BA individual. The estimated values agreed with the admixture proportions computed by the program qpAdm, using Yamnaya Samara and Anatolia (N) as ancestors, Yoruba, EHG and WHG as outgroups ($q_{steppe} = 53\%$, Fig. 4c). Considering a three-way admixture scenario, we included Serbia HG samples in the FA. For this analysis, the relative contribution of each gene pool to the other samples was measured by their coordinates in the triangle formed by the source populations (Fig. 4d). The ancestry proportions reflected in the FA plot were equal to (11%, 40%, 49%) from Serbia HG, Anatolia and Yamnaya, respectively, comparable to the values (8%, 47%, 45%) estimated by qpAdm. The samples from Neolithic Great Britain did not exhibit any shared ancestry with the Yamnaya source population, in agreement with negative values computed by qpAdm.

**Two-way analysis by regions**. To extend our focus on geographic regions, we considered FA of multiple populations from Bronze Age Germany and Czech Republic (66 samples), prehistoric and early Middle-Ages Scandinavia (28 samples)[38], Bronze Age and Roman Italy (17 samples)[39]. To compare with the results for Great Britain, we re-analyzed the samples from this region (66 samples). A two-source model of ancestry was assumed in each region, and the drift parameter was chosen to remove temporal effects from factor 2 (Supplementary Fig. 9). In samples from Great Britain, BA individuals shared closer relationships with samples from the Srubnaya culture than with samples from the Yamnaya culture (Fig. 5a). Samples from Neolithic Great Britain were close to Anatolian early farmers, with a relative position on the first factor suggesting a contribution from an unmeasured source of ancestry. In Germany and Czech Republic, Bell Beaker individuals exhibited levels of Steppe ancestry similar to those observed in Bell Beaker individuals from England (Fig. 5b). Samples from the Corded Ware culture and from the early Bronze Age were closer to the Yamnaya group than were the Bell Beaker samples. In Scandinavia, factor 1 represented the genetic ancestry of samples relative to early farmers and eastern hunter-gatherers from Latvia (Fig. 5c). The Yamnaya steppe samples were represented as a mixture of early farmers (55%) and eastern hunter-gatherers (45%), consistent with the admixture proportions from qpAdm (58% and 42% for the respective contributions of early farmers and Latvian HG). Like modern Finns (Supplementary Fig. 7), Swedish Vikings were closely related to the Yamnaya samples. In contrast, Swedish HG from Motala were closely related to the Latvian HG samples. In Italy, Langobard samples had levels of steppe ancestry similar to Bell Beakers, and those levels were lower than those observed in northern and central European populations (Fig. 5d).

**Comparative analyses of European samples**. To provide a comprehensive comparison of geometric representations and

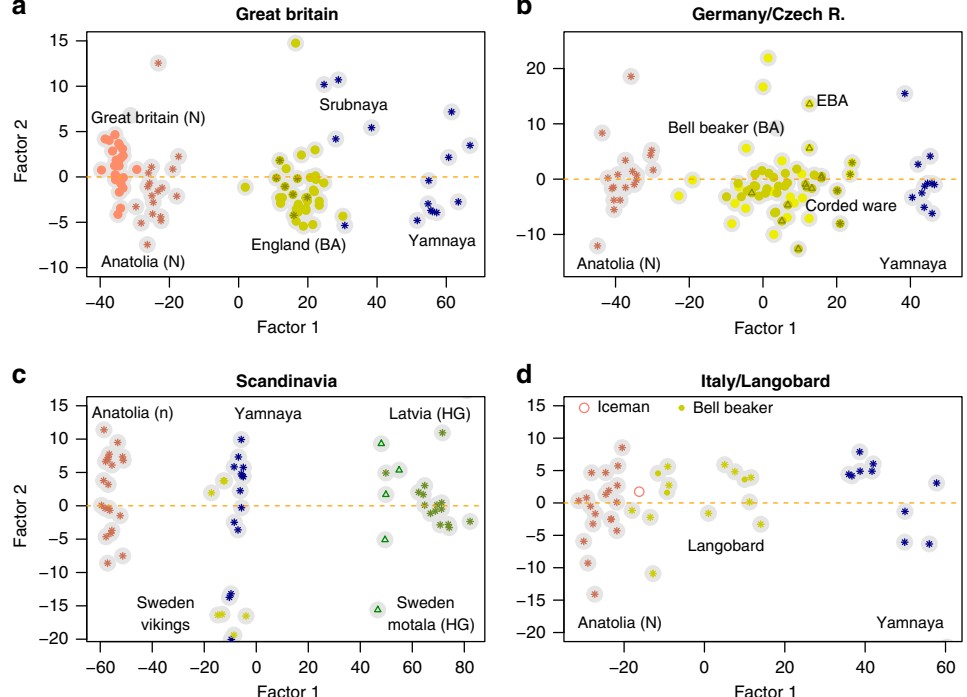

**Fig. 5 Factor analysis by regions.** Multi-population FA in four regions of Europe. **a** Bronze Age and Neolithic Great Britain, continued from Fig. 3, **b** Bronze Age Germany and Czech Republic (star: Corded Ware, triangle: EBA), **c** Prehistoric and early Middle-Ages Scandinavia (triangles represent Swedish HG from Motala), **d** Bronze Age and Roman Italy. HG: Hunter-Gatherers, N: Neolithic, BA: Bronze Age.

methods of analysis, we evaluated admixture coefficients for 22 ancient populations with four approaches. We compared ancestry coefficients computed from FA with admixture proportions computed by qpAdm, PC projections and STRUCTURE. In FA, PC projections and STRUCTURE, individual admixture estimates were averaged across individuals in population samples. Here ancestry coefficients computed from FA and PCA were used to decide whether or not the methods correctly represented ancestral relationships in geometric space.

In two-way admixture analyses, the estimates of FA and qpAdm strongly agreed with each other (Supplementary Fig. 10, Supplementary Fig. 11), providing evidence that positions on factor 1 corresponded with steppe ancestry. In FA, the proportion of variance explained by temporal distortions was non-negligible (mean: 17.7%, SD: 7.4%, Supplementary Table 1). Population samples from the Bell Beaker culture shared similar contributions from the Yamnaya gene pool in FA and in qpAdm, supporting a rapid spread of this culture during BA (Supplementary Fig. 12). Differences between FA and qpAdm were nevertheless observed for Corded Ware samples from Germany and Czech Republic which shared similar levels of steppe ancestries in FA (around 73–74%) and had different ancestries in qpAdm (87% and 55%). Samples from the neolithic period had no steppe component in FA, whereas qpAdm estimates were substantially larger than zero for some samples (England N, Scotland N, Germany LBK, Supplementary Fig. 10). Estimates from STRUCTURE correlated with those from qpAdm, but they exhibited a systematic bias toward higher proportions of Yamnaya ancestry. This bias could be explained by sensitivity to distortions similar to those observed in PCA (Supplementary Fig. 11). Estimates obtained from PC projections exhibited higher variance than those obtained from FA, and they were close to those computed with STRUCTURE (Supplementary Fig. 13).

In three-way analyses, ancestries were strongly correlated in FA and qpAdm, both for Yamnaya and for Anatolian sources (Fig. 6,

Figs. S14–S16). FA provided slightly higher values of steppe ancestry and lower values of Anatolian ancestry compared to qpAdm. The main differences between qpAdm and FA estimates concerned HG ancestry. Estimates of HG ancestry were consistent across samples for FA and for STRUCTURE, whereas they were variable across Corded Ware, Langobard, and early neolithic samples for qpAdm (Fig. 6, Supplementary Fig. 14). While STRUCTURE coefficients correlated with FA estimates, STRUCTURE over-evaluated Yamnaya ancestry in all samples (Supplementary Fig. 14). PC projections on present-day samples led to Anatolian ancestry similar to other programs, but failed to correctly represent the relative contributions of Yamnaya and HG to the samples, likely because the PC plots exhibited over-dispersion of samples along the Yamnaya – HG axis (Fig. 6, Figs. S14–S15). The over-dispersion phenomenon could explain the larger variance of ancestry estimates obtained from PC plots.

In conclusion, we introduced a FA method for describing ancestral relationships among DNA samples collected at distinct time points in the past. Using the Brownian model of allele frequency drift, we compared the results of FA with those of frequently-used methods, PCA, STRUCTURE, and F-statistics. The results suggested that the geometric representations of samples provided in FA plots were accurate, and the distances between samples agreed with ancestry estimates derived from F-statistics. Representations based on PC projections were less accurate than FA due to large variance along some axes. Temporal distortions affected STRUCTURE in a way similar to PCA, and correcting for drift revealed hidden population structure better than did PCA or current ancestry estimation approaches.

In a re-analysis of a merged data set of ancient DNA filtering out SNPs with high levels of missing data and genomes of low coverage, we implemented correction for temporal drift to describe ancestry in samples from ancient Europeans and Eurasians. After correction, the patterns observed in FA plots

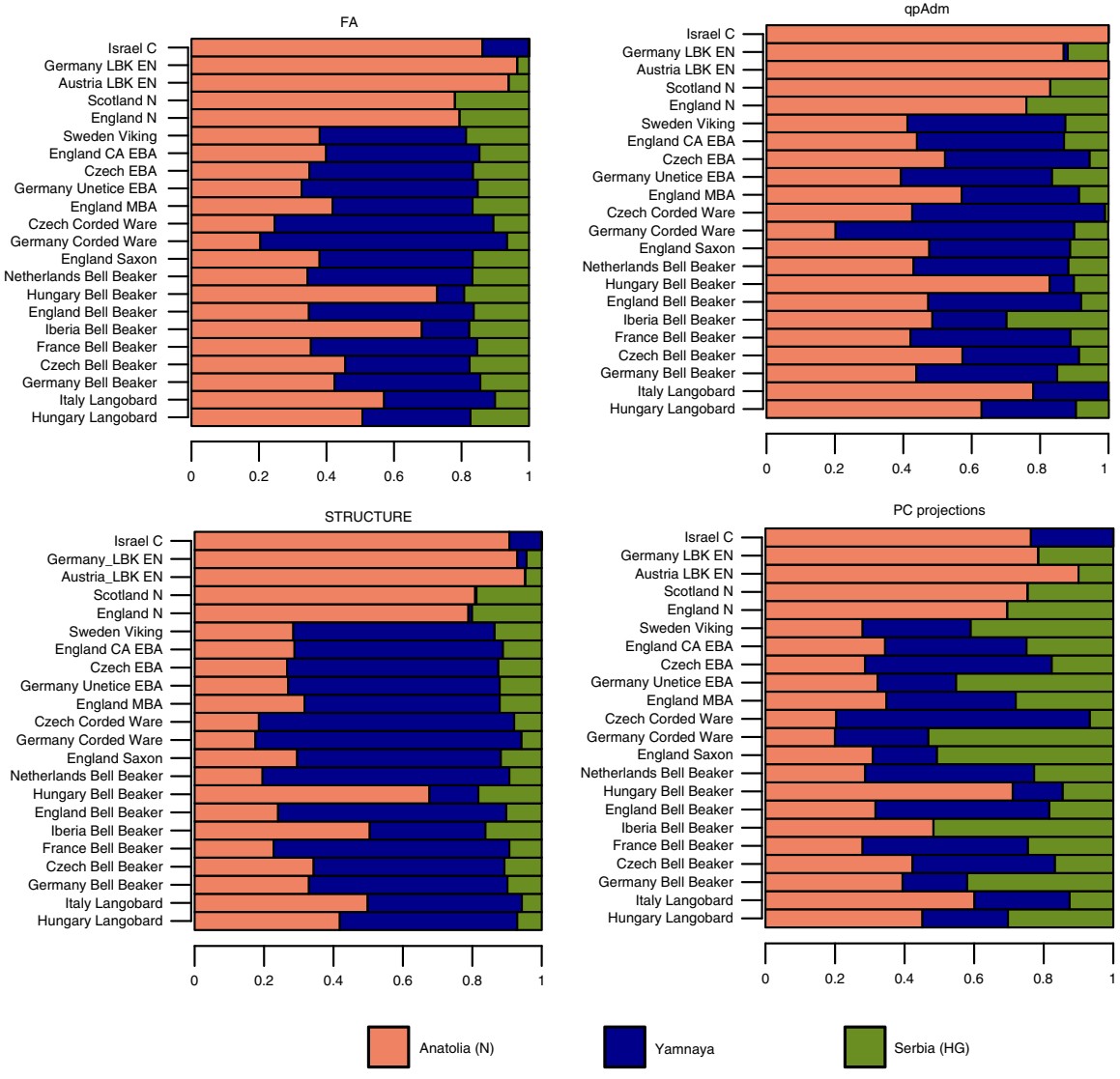

**Fig. 6 Three-way admixture coefficients for 22 ancient population samples.** Ancestry coefficients computed using Anatolia (N), Yamnaya Samara, and Serbia Iron Gates (HG) as ancestral populations. FA: factor analysis with $\lambda = 0.05$, qpAdm: using Yoruba, Russia Sidelkino (EHG), France-Ranchot (WHG), Aleut and Altaian samples as outgroups, STRUCTURE: implemented with sparse non-negative matrix factorization, PC projections: computed with present-day European samples from the 1000 Genomes Project. HG: Hunter-Gatherer, N: Neolithic, BA: Bronze Age, CA: Copper Age, E: Early, M: Middle.

provided a better representation of samples than those observed in projections of ancient samples on axes built on present-day European genomes. The FA supported the hypothesis that a major change in genetic mixture of individuals occurred in Great Britain and in continental populations around 4500 years BP[4]. Our analysis provided a geometric representation of samples consistent with the proportions of steppe, early farmer and hunter-gatherer ancestries estimated with *F*-statistics. Those results contrasted with PCA results which were obscured by statistical errors. They also differed with admixture estimates obtained with STRUCTURE which led to a systematic bias toward steppe ancestry in European samples. For all ancient European populations, FA provided admixture estimates that generally agreed with *F*-statistics, and were more accurate for small or large coefficients. Including corrections for temporal drift resulted in an algorithm with a computational cost similar to a PCA. A useful and important feature of our approach is to avoid supervised analyses in which present-day populations may be over-represented, or for which specific outgroup populations must be considered. The unsupervised approach based on FA

revealed details of population structure masked in PC projections, and was generally more accurate than other methods of inference of population structure for ancient samples.

## Methods

**Details on factor analysis**. Let **Y** be an $n \times p$ matrix of genotypic data, where $n$ is the number of individual samples and $p$ is the number of markers, typically represented as SNPs. We suppose that the data are centered, so that the mean value for each sample is null. We also suppose that each sample, $i$, is associated with a sampling date, $t_p$, corresponding to the age of the sample. In our model, the date $t_1$ corresponds to the most ancient sample while $t_n$ corresponds to the most recent sample, and the dates were transformed to fall the unit interval $0 \le t_1 \le \cdots \le t_n \le 1$ as follows. Let $a_i$ be the age of sample $i$, for example measured in years cal BP. First, the transformation converts ages into forward values by setting $s_i = 1 - (a_i - \min a_j)/(\max a_j - \min a_j)$. Then, it computes the variance of each row of **Y** as $v_i = \text{var}[Y_{i,}]$, and sets $t_i = \min v_j + s_i(\max v_j - \min v_j)$. With this transformation, the sample times vary within the range of allele frequency variances.

Our FA model takes the following form, $\mathbf{Y} = \mathbf{U}\mathbf{V}^T + \epsilon$, where **U** is an $n \times K$ matrix of scores, and $\mathbf{V}^T$ is a $K \times p$ matrix of loadings. To incorporate correction for temporal drift, we model the error term, $\epsilon$, as a multidimensional Gaussian distribution with mean 0 and covariance matrix $\alpha\mathbf{C} + \sigma^2\mathbf{I}$, where $1/\alpha$ is a scale parameter for temporal drift, $\sigma^2$ is the variance of residual errors, and **I** is the $n \times n$ identity matrix. In this model, a key parameter is the ratio $\lambda = \sigma^2/\alpha$, which

measures the relative importance of temporal drift—influenced by effective population size and divergence or admixture times—with respect to the residual variance $\sigma^2$—reflecting model approximation and sampling error.

To model the effect of temporal drift, we assumed that the $n \times n$ covariance matrix $\mathbf{C}$ is given by the Brownian process, $c_{ij} = \min(t_i, t_j)$ for all pairs of samples $i, j$ in $1, \ldots, n$[42,43]. As a consequence of this definition, the variance in allele frequencies is proportional to time. To compute the factor matrix, $\mathbf{U}$, we turned to an equivalent formulation of the factor model as

$$\mathbf{Y} = \mathbf{W} + \mathbf{Z}\mathbf{B}^T + \epsilon', \qquad (1)$$

where the residual noise is now described by $\epsilon' \sim \mathbf{N}(0, \sigma^2\mathbf{I})$. The above formula introduces correction factors, $\mathbf{Z}$, that are computed after a spectral decomposition of the covariance matrix $\mathbf{Z} = \mathbf{P}\sqrt{\Lambda}$, where $\mathbf{P}$ is a unitary matrix of eigenvectors, and $\Lambda$ is the diagonal matrix containing the eigenvalues of $\mathbf{C}$. In our FA algorithm, the correction factors were computed with a standard linear algebra library in R (Supplementary Note 1).

The FA model considers an $n \times p$ matrix of effect sizes, $\mathbf{B}$, with coefficients defined as i.i.d. random variables with Gaussian prior distribution, $N(0, \alpha)$. The latent matrix, $\mathbf{W}$, representing the data matrix after its correction for temporal drift, has an uninformative prior distribution. Statistical estimates for the matrices $\mathbf{W}$ and $\mathbf{B}$ are obtained by maximizing a posterior distribution in this Bayesian framework. This approach amounts to finding the minimum of the following loss function

$$\mathcal{L}(\mathbf{W}, \mathbf{B}) = \frac{1}{2} \| \mathbf{Y} - \mathbf{W} - \mathbf{Z}\mathbf{B}^T \|_F^2 + \frac{1}{2}\lambda \| \mathbf{B} \|^2 , \qquad (2)$$

where $\lambda = \sigma^2/\alpha$. The estimate of $\mathbf{W}$ represents the best approximation of rank $K$ of the matrix $\mathbf{Y}$, with respect to the following matrix norm

$$\|\mathbf{Y}\|_A^2 = \mathrm{Tr}(\mathbf{Y}^T\mathbf{A}\mathbf{Y}) , \qquad (3)$$

with $\mathbf{A} = \mathbf{P}\mathbf{D}_\lambda\mathbf{P}^T$ and $\mathbf{D}_\lambda$ is a diagonal matrix with coefficients

$$\mathbf{D}_\lambda(i,i) = \frac{\lambda}{\lambda + \lambda_i} , \quad i = 1, \ldots, n. \qquad (4)$$

Following[33], the optimal solution is equal to $\mathbf{W} = \mathbf{P}\mathbf{D}_\lambda^{-1/2}\mathrm{svd}_K(\sqrt{\mathbf{D}_\lambda}\mathbf{P}^T\mathbf{Y})$. The $K$ corrected factors forming $\mathbf{U}$ and their associated loadings, $\mathbf{V}$, can then be obtained from the SVD of the matrix $\mathbf{W}$. The complexity of the algorithm is of the same order as the SVD algorithm, $O(npK)$, and allows fast implementations on standard computers. Because FA is defined as an unsupervised regression model, the limitations of FA in terms of sample size are similar to those of linear regression models, ANOVA models or PCA. For the choice of the number of factors, we used $K = 2$ factors in two-way admixture analyses and $K = 3$ factors in three-way admixture analyses. For the choice of the drift parameter, we chose the largest value of $\lambda$ that removed the effect of time in the $K$th factor, where $K$ was the number of putative ancestral populations (Supplementary Fig. 9).

**Evidence of distortions in PCA.** We used the computer program *msprime* version 0.7.6 for Linux to simulate samples for individuals at distinct time points in the past in neutral and divergence coalescent simulations[44]. Firstly, a single population of $N_e = 10{,}000$ individuals was simulated during 4000 generations. An individual was sampled every 100 generations, resulting in 41 samples with ages ranging between 0 (present-day) and 4000 generations. A total of around 9000 SNPs were simulated for each individual. Secondly, a divergence model was considered in which an ancestral population of effective size $N_e = 10{,}000$ split into two sister populations of equal sizes 1500 generations ago. Twenty-four individuals with ages ranging from 0 to 1000 generations were sampled every 100 generations, and around 8800 SNPs were simulated for each individual. One hundred replicates were created for the divergence scenario. For each replicate, we computed the Davies-Bouldin index evaluating the degree of clustering in multidimensional data[45]. Corrections for temporal drift in allele frequency were expected to provide values closer to one than those computed from uncorrected components.

**Evidence of shrinkage in PC projections.** Coalescent simulations of admixture models were considered in which an ancestral population of effective size $N_e = 10{,}000$ split into two sister populations of equal sizes 1300 generations ago. The ancestral populations came into contact 800 generations later, and this event gave rise to an admixed population. Individuals in the admixed population shared 75% of their ancestry with the first ancestral population, and 25% of their ancestry with the second ancestral population. One hundred individuals were sampled from the present-day admixed population. Fifty individuals were sampled from each ancestral population, one thousand generations ago. A total of around 9600 SNPs were simulated for each individual. One hundred replicate data sets were created for this scenario. To evaluate the statistical errors of admixture estimates, we computed mean squared errors (MSE) both for PC projections and for FA.

**Ancestry estimates from FA and PCA.** In two-way admixture models, estimates of ancestry coefficients were computed based on the relative positions of cluster means along the first axis, $\mathbf{u}$. For estimating individual admixture estimates from two source populations $a$ and $b$, the centers of each source population, $c_a$, $c_b$ were

first estimated, and the ancestry coefficient from source $a$ was computed of each individual $i$ from its first factor coordinate, $u_i$, as $q_{ai} = (c_b - u_i)/(c_b - c_a)$. The ancestry coefficients were then constrained to take their values between 0 and 1 by truncation, and averaged over admixed individuals to provide a population estimate. Mean squared errors for estimates obtained from FA and from projections on present-day PCs were then computed. With more than two source populations, ancestry estimates were evaluated by computing the coordinates of each sample in the coordinate system formed by the sources. The coordinates are non-negative numbers when the target samples lie inside the convex set formed by the ancestral population centers. Negative coordinates are indicative of no contribution from the corresponding source.

**Generative model simulations.** We generated genotypes for ancestral and admixed individuals by using factor models with a probit link function. Factor models are generative models close to PCA and STRUCTURE[9,25]. While matrix factorization methods underlie inference in STRUCTURE and PCA, these methods can also be used to generate new data. Considering two ancestral populations, the relative distance between population centers on the first factor axis was proportional to the square root of the divergence time[12]. Samples from the admixed population were generated at distance between the population centers proportional to the ancestry shared with each source. The generative model incorporated a time frame for simulating samples at distinct time points based on Eq. (1). In those simulations, the timing of the admixture event and the effective population sizes were reflected into the drift coefficient, $\lambda$ (or equivalently, $\alpha$). As with coalescent simulations, a first scenario considered a divergence model in which two populations evolved without gene flow. In this case, the populations clustered separately along the first factor. The samples were taken at random time points in the past. A second scenario considered an admixture model in which two populations diverged in the distant past and an admixture event occurred recently. Half of the samples were ancient, taken from the ancestral populations at random time points in the past, and the other half of the samples were collected from the admixed population in present time. The number of samples, $n$, was equal to 200, and the number of markers was kept to $p = 5000$. We performed 100 simulations for each scenario.

For the divergence model, the factor matrix $\mathbf{U}$ contained $K = 3$ factors, simulated as Gaussian random variables. The standard deviation for the first factor, $s_1$, measuring the divergence between the two ancestral populations, was varied in the range $(2,10)$, which corresponded to an $F_{ST}$ in the range $(0.02, 0.25)$. Factors 2 and 3 had lower standard deviations than factor 1, respectively equal to $s_2 = 1.5$ and $s_3 = 0.5$. The $\lambda$ parameter, reflecting the amount of temporal drift, was chosen in the range $[10^{-1}, 10^{-6}]$ representing weak to strong drift intensities. Loadings were simulated as standard Gaussian random variables, and the residual variance was $\sigma^2 = 1$.

For the admixture model, the two ancestral populations were positioned with a distance separating their centers varied in the interval $[10, 12]$, corresponding to an $F_{ST}$ around 25%. Factors 2 and 3 had standard deviations equal to $s_2 = 1.2$ and $s_3 = 1$. Admixed individuals were positioned so that the center of the admixed population was at relative distance $a$ from ancestral population 1, and $1 - a$ from ancestral population 2, where $a$ represents the ancestry contribution of population 1 to present-day samples. The ancestry coefficients ranged between $a = 0.2$ and $a = 0.4$. The $\lambda$ parameter was set to $\lambda = 5.10^{-2}$ (strong drift effect). The number of samples was set to $n = 200$, and the number of markers was kept to $p = 5000$. The loadings were simulated as independent Gaussian random variables, $N(0,0.2)$, and the residual error was set to $\sigma = 0.1$.

**Benchmarking representations by comparing ancestry estimates.** We used population ancestry estimates obtained with the method described above to verify that the relative distances between samples in FA plots accurately reflected ancestral relationships, and to evaluate whether the geometric representations were consistent with ancestry estimates provided by other methods. In these plots, distances between ancestral populations and their descendant populations should reflect the amount of ancestry contributed by each gene pool. In our experiments, we compared ancestry estimates obtained from FA with estimates obtained with the program qpAdm[2,8,27], with STRUCTURE estimates obtained with the snmf function from R package LEA 2.4[46], and with estimates from PC projections on admixed samples. Ancestry analyses using qpAdm were realized in the R package admixr (version 0.7.1), which calls the ADMIXTOOLS program (version 6.0 for macOSX and Linux)[8,47]. The qpAdm estimates are based on models of population genealogies and account for temporal drift by computing branch length correlations. The snmf algorithm implements an accelerated version of the ADMIXTURE program with accuracy similar to ADMIXTURE[24,26]. The principle of the snmf algorithm relies on sparse non-negative matrix factorization, and is rooted in the equivalence between PCA and STRUCTURE.

We performed generative simulations of samples with admixture from two sources, including ancient samples from each source. We varied the level of divergence between the sources, measured by $F_{ST}$, the level of admixture in modern samples, and the drift coefficient. We sampled around 9k SNPs for $n = 200$ individuals, including 100 admixed samples and 50 samples from each ancestral source. Two levels of divergence between ancestral populations were considered, corresponding to an $F_{ST} \approx 5\%$ and an $F_{ST} \approx 25\%$. The admixture coefficient was varied in the range $[0.05, 0.5]$. The drift coefficient, $\lambda$, took three values equal to

$\lambda = (5 \times 10^{-2}, 1.5 \times 10^{-2}, 5 \times 10^{-3})$ for the higher level of divergence, and to $\lambda = (10^{-1}, 5 \times 10^{-2}, 10^{-2})$ for the lower level of divergence. Those values were referred to as weak, medium, and strong drift in each case. Statistical errors were computed with respect to a ground truth estimate of the admixture coefficient obtained from the exact representation of the samples in the two axes created by the generative model. Estimates from STRUCTURE (snmf) and from PC projections were expected to be close to qpAdm estimates when the amount of drift is small, for example, for recent admixture in large sized populations, and less accurate otherwise due to variance caused by shrinkage and to bias caused by temporal distortions.

**Ancient human DNA samples**. A merged data set consisting of genotypes for 5081 ancient and present-day individuals compiled from published papers was downloaded from David Reich's lab repository (reich.hms.harvard.edu). The downloaded data matrix contained up to 1.23 million positions in the genome. Considering age defined as average of 95.4% date with range in cal BP computed as 1950 CE, Eurasian samples with age less than 13,980 years were retained. The data matrix was filtered out for samples having a minimum coverage of 0.25x. The resulting ancient samples had a median coverage of 2.17x (mean of 2.88x) and a maximum of 25.32x. Genomic positions with less than 34% of missing genotypes were analyzed. Missing genotypes were imputed by using a matrix completion algorithm based on sparse non-negative matrix factorization[26].

Prior to PC and FA analysis, we adjusted the centered matrix of genotypes for coverage by performing a regression analysis of the form $Y_{ij} = a_j \text{coverage}_i + \tilde{Y}_{ij} + e_{ij}$. This regression analysis assumed that lower coverage values could correlate with particular population samples. Corrections were performed by fitting a latent factor mixed model with ten latent factors and coverage as a primary variable (R package lfmm version 1.0[33]). In subsequent analyses, the genotype matrix was replaced by the latent matrix, $(\tilde{Y}_{ij})$, adjusted in the regression model. After filtering individuals and genotypes, the resulting data contained 143,081 genotypes for 521 present-day European individuals from the 1k Genomes project and 704 ancient Eurasian samples from previous studies[40,41] Sample IDs and associated metadata were included in Supplementary Data 1. The most important contributions to the samples included in our data set were (1) 201 ancient individuals in ref. [4], (2) 144 ancient individuals in ref. [37], (3) 79 ancient individuals in ref. [3] (38 same samples in ref. [2]), (4) 56 ancient individuals in ref. [36], and (5) 50 ancient individuals in ref. [38].

In applying FA to the merged data, the drift parameter was chosen in order to remove the effect of time on the Kth factor, where K is the number of ancestral groups considered. The STRUCTURE model was implemented with snmf using a regularization parameter $\alpha = 1,000$[26]. Present-day samples for PC projections included populations from The 1000 Genomes project, mainly from Great Britain, Italy, Spain, Finland and Russia including Siberia (Supplementary File 1). Two-way admixture models were implemented in the program qpAdm by using samples from Anatolia (N) and Russia (Yamnaya, Samara) as ancestral groups, and Yoruba (YRI), Russia Sidelkino (EHG), and France-Ranchot (WHG) samples as outgroups. Three-way admixture models were implemented in the program qpAdm by using samples from Anatolia (N), Russia (Yamnaya, Samara), and Serbia (Iron Gates, HG) as ancestral groups, and Yoruba (YRI), Altaian, Aleut, Eskimo Chaplin, Russia Sidelkino (EHG), France-Ranchot (WHG) samples as outgroups. Regarding sample ages, we verified the robustness of our analyses to uncertainty in sample ages, by replicating analyses with random ages drawn from reported confidence intervals. The confidence intervals spanned approximately 400 years, and the ages were randomly drawn around the central estimate with a standard deviation of 100 years. We created 100 replicates for each data set in two-way admixture analyses of European populations. Sampling individual ages from their confidence intervals biased the admixture coefficients toward intermediate values (50% Yamnaya-50% Anatolia). The bias was around 1%, and it remained moderate for all studied populations. The projections of individuals on factor 1 were highly stable (S.D. ranged between 0 and 1.3%), and the squared correlation of the replicated factor 1 with the factor obtained using the central estimate was around 99.9%. These results showed that, for ancient European samples, FA results were robust to moderate levels of uncertainty in sample date estimates (Supplementary Table 2). Analyses were performed using R 3.6.3 for OS X 10.11, and R 3.6.4 for Linux[34].

**Reporting summary**. Further information on research design is available in the Nature Research Reporting Summary linked to this article.

## Data availability

Data were extracted from a public data set consisting of genotypes for 5081 ancient and present-day individuals compiled from published papers and available from David Reich's lab repository (February 2019, v.37.2): reich.hms.harvard.edu. All data have already been published, and have permissions appropriate for fully public data release. The IDs of samples included in our study were provided in Supplementary Data 1, and the corresponding DOIs were referenced from the database.

## Code availability

The methods described in this manuscript were implemented in the R package tfa available at: bcm-uga.github.io/tfa/ under a GNU General Public License v3.0. A user guide and a tutorial are available at: bcm-uga.github.io/tfa/articles/tfa-vignette.html. Computer codes for simulation and data analyses were made available from a GitHub repository github.com/francoio/Francois_and_Jay_2020 under a GNU General Public License v3.0.

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

## Acknowledgements
F.J. thanks the Paris-Saclay Center for Data Science 2.0 (IRS) for supporting her mobility. O.F. thanks the Grenoble-Alpes Data Institute, supported by the French National Research Agency under the Investissements d'Avenir program (grant number ANR-15-IDEX-02). J.F. and O.F. are grateful to Cyril Furtlehner and Céline Bon for stimulating discussions, and to Benjamin Demaille and Séverine Liégeois for their help with simulations at early stages of method development.

## Author contributions
O.F. and F.J. conceived the study, developed the method, carried out analyses, and wrote the manuscript.

## Competing interests
The authors declare no competing interests.
