## [Peer Review File · Nature Communications]

REVIEWER COMMENTS

Reviewer #1 (Remarks to the Author):

The authors have developed a new method to visualize the relationship between ancient and present-day samples using factor analysis. This method, compared to principal component analysis, has the advantage of taking into account differences in the amount of genetic drift between samples due to changes in the age of the fossils analyzed. The authors show that the age of the fossils analyzed can produce spurious sinusoidal shapes in principal component analysis, and that the effect can be corrected using factor analysis. The authors show that factor analysis produces a more accurate estimate of the proportion of ancestry coming from a set of populations compared to estimates obtained using principal component analysis, qpadm and structure. Then, the authors show an application of their method to ancient European samples.

Principal component analysis has been widely applied to studies of ancient DNA, and the community has been aware of its limitations when studying temporal samples. Factor analysis is a method that more accurately represents the relationship between present day and ancient samples compared to principal component analysis. Due to this, the method proposed by the authors is a very important contribution to the field of ancient DNA studies and I predict that it will be widely used. The majority of the comments that I have on the manuscript are related to the clarification on concepts. After they are corrected, this manuscript will be suitable for publication.

Major details

Please provide more details on the running time of the method. How long will the method take to run as a function of the number of samples and SNPs using one particular CPU? The only mention of the performance of the method is on line 520, page 33 using a sample size of 1,300 and 150,000 SNPs.

The authors might want to consider including comparisons with DyStruct (Joseph and Pe'er, 2019, AJHG) in their analysis on Figure 2. DyStruct is a model-based approach for inferring shared ancestry when including temporal samples. DyStruct is designed to work with ancient DNA and outperforms ADMIXTURE (and therefore STRUCTURE) on analysis of ancestry estimates. It would be good to see if DyStruct works as well as factor analysis on analysis of ancestry estimates.

Figure 1A) I am having some problems interpreting this plot. In the label of the Figure the authors mention: "Samples with ages ranging from 0 (present-day, grey color) to 4,000 generations (past, dark blue color) were simulated. A) Covariance matrix for observed samples," If I am interpreting Figure 1A) correctly, the colors for each square shown in the covariance matrix are not related to the ages of the sample, but to the covariance between a pair of samples. What is the interpretation that should be given to each color? Is the color related to the amount of covariance between a pair of samples? Also, should I interpret that there is a low covariance between samples taken at 4 ky and all of the samples because there is a predominant blue color in the comparisons between samples taken at all ages and the sample taken at 4ky?

Page 7 Line 126. "The patterns observed in the sample covariance matrix were highly similar to those obtained in a theoretical covariance model corresponding to a Brownian process" Please expand more on the methods on how you obtained the theoretical covariance model corresponding to a Brownian process.

Page 10 Line 167. "we performed simulations of ancestral and admixed genotypes based on latent factor models." Please provide more details on methods on how this is done.

Figure 1D) The authors mention that "In a PCA of simulated samples, PC1 reflected the level of divergence between populations" Am I correct in interpreting that the samples with a PC1 value

smaller than 0 belong to one particular population while the samples with a PC1 value bigger than 0 belong to the other population? If that is the case, it would be helpful to label the samples with a different shape. As an example, the samples in the left side of the plot could be labeled with a star while the samples in the right side could continue to be labeled with the circle.

Page 40 Line 672 "Prior to PC and FA analysis, the matrix of genotypes was corrected for the effect of genomic coverage by fitting a latent factor mixed model with coverage considered as the primary variable (Caye et al. 2019)." Please provide more details on how this is done.

Provide a little bit more of information on the README.md file from <https://github.com/bcm-uga/tfa> on how to run the methods. I recommend to list what methods are contained in the repository and giving a small manual on how to run them.

Minor details

Figure 1A) The Y-axis label should indicate the sample age in ky.

Page 4 Line 64.- "In this study, we provide evidence that representations of ancient DNA samples based on PC projections, STRUCTURE or PCA disagree with each other and with F-statistics. We introduce a factor analysis that enables geometric representations of ancient samples in better agreement with the later" Does 'later' refer to F-statistics here?

Page 12 Line 185 "Considering a lower level of divergence between ancestral populations did not change the ranking of method performances (Figure S3)." What is the level of divergence on Figure S3 compared to Figure 2?

Page 12 Line 187 "When the drift parameter decreased, estimates from PC projections exhibited less bias, and became closer to qpAdm values (Figure S3B)." It is hard to observe this from Figure S3B. If the authors wish to make this point clearer, it would be good to make an additional boxplot (or a similar plot) that proves this claim.

Page 12. Line 190 "We eventually performed "realistic" simulations with low differentiation, low drift, and relatively recent mixture consistent with the demography of ancient Europeans (Figure S4)". I recommend providing a figure in S4, or at least an explanation in the caption, of the scenario you are simulating that is consistent with the demography of ancient Europeans.

Page 17 Line 252 "A two-source model of ancestry was assumed in each region, and the drift parameter was chosen to remove temporal effects from factor 2 (Figure S9)." Please expand on how this is done on the caption of Figure S9.

Page 30 Line 464.- "The dates are normalized to fall the unit interval $0 < t_1 \leq \dots \leq t_n \leq 1$." I believe the inequality should be equal to $0 \leq t_1$.

Page 35 Line 561.- "we generated genotypes for ancestral and admixed individuals from a factor model with a probit link function, providing simulations for which a ground truth was available for the comparisons of methods." How does this work? Please provide more details.

Reviewer #2 (Remarks to the Author):

François and Jay present a very useful new method for controlling for the effects of temporal drift when performing exploratory data analysis of population structure on ancient genomes. I think the methodology is clearly laid out, the figures are beautiful and I appreciate that the code is made readily available for potential users. I would recommend this paper for acceptance with only a few minor edits. I mostly have comments and suggestions for text clarifications. I'm happy if the

authors think certain analyses might require lots of work and should be left to a future study.

How does your method perform if the ages are misspecified? In other words, how robust is it to mis-dated samples at different levels of mis-dating? How did you deal with ranges of C14 dates? Did you take their midpoint?

Is the utility of the method dependent on sample size? You apply this to a dataset of hundreds of human genomes. If you were to apply it to a dataset of a few genomes (e.g. by subsampling your current data), would you perform as well as PCA, better or worse?

Could you motivate a bit more why agreement with F-statistics / qpAdm should be seen as the "gold standard"?

Can you elaborate on similarities / differences between this model and the LFMM model for environmental association? It seems like this is similar to the models in Frichot et al. and Caye et al.?

Is there information in the matrix B about SNPs that vary strongly with time, and could potentially be used to detect on individual variants over time?

In Figures 1A and 1C, what do you mean in the y-axis label when you say "Samples"? As this is a covariance matrix, I'm guessing this should be the same label as the x-axis, and should say "Sample age (ky)" as well? Can you clarify in the caption to Figure 1C which matrix this is in your model? I'm guessing matrix C but it doesn't seem to represent $\min(t_i, t_j)$ in each of its cells, so I'm a bit confused as to what this represents.

In some sense, the temporal drift you're correcting away is also an important feature of the data. Could estimates of temporal drift be represented in separate plots when studying real data, along with the factor plots?

Could the estimates of temporal drift be used as a measure of population size changes over time? (essentially looking at how the temporal variance in allele frequencies for a SNP relate to the observed dates)

Line 39, page 3: "counterpart" should be plural

Reviewer #1:

The authors have developed a new method to visualize the relationship between ancient and present-day samples using factor analysis. This method, compared to principal component analysis, has the advantage of taking into account differences in the amount of genetic drift between samples due to changes in the age of the fossils analyzed. The authors show that the age of the fossils analyzed can produce spurious sinusoidal shapes in principal component analysis, and that the effect can be corrected using factor analysis. The authors show that factor analysis produces a more accurate estimate of the proportion of ancestry coming from a set of populations compared to estimates obtained using principal component analysis, qpadm and structure. Then, the authors show an application of their method to ancient European samples.

Principal component analysis has been widely applied to studies of ancient DNA, and the community has been aware of its limitations when studying temporal samples. Factor analysis is a method that more accurately represents the relationship between present day and ancient samples compared to principal component analysis. Due to this, the method proposed by the authors is a very important contribution to the field of ancient DNA studies and I predict that it will be widely used. The majority of the comments that I have on the manuscript are related to the clarification on concepts. After they are corrected, this manuscript will be suitable for publication.

We thank you for positive comments and for suggesting some clarifications of concepts. Some of these concepts were new, and we agree that they require careful explanations. Here is a summary of the changes we made to the manuscript to account for your suggestions.

We performed simulations of running times (Q1). In agreement with complexity estimates, we found that the running times for FA were approximately twice those of PCA. We think that this information is more interesting than including running times for a particular CPU, because it does not depend on the computer or the programmer. We briefly summarised the theoretical and experimental results in the text.

We performed a preliminary comparative analysis of “dystruct” performances (Q2). In addition to their size, the data considered in our manuscript use many more time-points than “dystruct” could reasonably handle. In our preliminary analysis, dystruct was much slower or much more energy consuming than the other methods considered, requiring hours/days compared to a few seconds/minutes for PCA, FA or snmf. We abandoned “dystruct” analyses because we realised that including them would not clarify our main message, and might even modify it by giving unbalanced analyses of ancestry estimation methods vs descriptive methods.

We clarified the relationships between the Brownian covariance model and the empirical covariance of allele frequencies under genetic drift (Q3-Q4). We also clarified the links between factor models and classical population genetic concepts such as drift, differentiation statistics, admixture, in simulation models and in data analyses (Q5-Q6).

We explained how correction for coverage was performed, and we modified our computer program to include additional functionalities that enable several methods to be easily reproduced (adjusting for coverage, choosing the drift parameter, computing ancestry estimates, computing the proportion of variance explained by the covariance model) (Q7-Q8). We provided detailed answers to all comments considering them as important remarks.

Q1. Please provide more details on the running time of the method. How long will the method take to run as a function of the number of samples and SNPs using one particular CPU? The only mention of the performance of the method is on line 520, page 33 using a sample size of 1,300 and 150,000 SNPs.

Running times strongly depend on the way the algorithm is implemented in a computer program. To avoid dependencies of our program on external libraries, our choice was to use the SVD function from the base package of the R statistical program

<https://www.rdocumentation.org/packages/base/versions/3.6.2/topics/svd>

Unlike running times, complexity analyses are not dependent on the CPU. Since the FA algorithm is exact (and not iterative), we could estimate the complexity of the algorithm to be of order $O(np)$, n the number of individuals and p the number of variants. A pseudo-code for the algorithm is provided in Table S1. Running times can be understood from the description of the algorithm, which requires one SVD to return the latent matrix. A second SVD is then needed to compute the factors from the latent matrix.

With our choice for SVD, the running times were around twice the running times of a classical PCA computed from the base SVD (this could be accelerated). As suggested, we measured how long the program took to run for $n = 100-300$ samples and $p = 2k-100k$ SNPs using a single CPU (Processor 2.5 GHz Intel Core i7 in MacBook Pro Laptop), and we compared the results with PCA. Note that we used a very common CPU, not optimised for matrix calculations.

These results are presented below. The factor 2 was confirmed. Because we did not optimize the code by using low-rank SVD, the results remained the same for higher or lower values of K . Note that the running times are much shorter than those for ADMIXTURE and other iterative algorithms requiring multiple runs.

We modified the manuscript Line 122 Page 7.

“For p much larger than n , the complexity of the algorithm is of order $O(npK)$. The running times can be understood from the description of the algorithm in Table S1 which requires one SVD to estimate the latent matrix. A second application of the SVD is then needed to compute the factors from the latent matrix. In experiments using the SVD algorithm implemented in the base library of the R programming language (R Core Team, 2018), the running times for FA were around twice the running times of a classical PCA.”

R Core Team (2018). R: A language and environment for statistical computing. R Foundation for Statistical Computing, Vienna, Austria. URL <https://www.R-project.org/>.

Caption: Running times of FA compared to PCA measured in seconds. Number of individuals (n) in the range 100-300, number of variants (p) in the range 2.5k-100k. The number of factors was set $K = 3$ in two-way admixture simulations. Processor: 2.5 GHz Intel Core i7 in MacBook Pro Laptop. The running times for FA are approximately twice longer than the running times for PCA and consistent with the $O(np)$ estimates of algorithmic complexity for both algorithms. The dashed blue line is $y=2x$.

Q2. The authors might want to consider including comparisons with DyStruct (Joseph and Pe'er, 2019, AJHG) in their analysis on Figure 2. DyStruct is a model-based approach for inferring shared ancestry when including temporal samples. DyStruct is designed to work with ancient DNA and outperforms ADMIXTURE (and therefore STRUCTURE) on analysis of ancestry estimates. It would be good to see if DyStruct works as well as factor analysis on analysis of ancestry estimates.

The primary objective of our method is to enable geometric representations of ancient samples (descriptive method). In contrast dystruct and other programs like structure, admixture, snmf, qpAdm focus on providing the estimates of ancestry coefficients for ancient and present-day samples. Of course, these methods provide graphical outputs (barplots), but not directly comparable to PC plots.

Our comparison study included two factor methods (PCA, FA) and two ancestry estimation programs (Structure/snmf, qpAdm). We believe that adding dystruct would provide an unbalanced representation of ancestry estimation programs vs descriptive methods, and take readers away from the main message of the manuscript.

We feel that qpAdm fills the objective of providing ancestry coefficients corrected for drift at least at a population level, and it is the state-of-the-art method for ancient samples (Harney et al. 2020). Using dystruct would at best duplicate qpAdm analyses, and overweight the importance of ancestry estimation methods (vs geometric representation methods). We used snmf/structure because they

are representative of ancestry estimation algorithms devised to study present-day samples (snmf is much faster than admixture or structure).

We also feel that comparisons between qpAdm, dystruct, and ancestry estimates derived from FA are interesting, but they deserve a separate study, which will be complicated by the very long running times of dystruct compared to the other methods. To illustrate the challenge, we performed a preliminary analysis comparing dystruct, FA and structure/snmf on relatively small datasets with 100 individuals and 2000 loci in two-way admixture models. The admixture rates ranged from 10%, to 45%. In this preliminary analysis, dystruct overestimated admixture coefficients in all cases, while Structure/snmf overestimated the coefficients in case of low admixture only. FA performed better than dystruct and structure. FA and snmf running times were similar (~0.1 second for one dataset on a single CPU), whereas dystruct running times were 25,000 times longer (50 minutes on 1 cpu). Comparing the three methods under all simulated and real data sets considered in our study would take a considerable amount of computational power in addition to shifting the scope of the paper.

Caption: Estimates of population admixture coefficients from simulated data. Eight genotypic matrices have been simulated for two-way admixture models (F_{st} between source around 5%, 100 individuals, 2000 loci). Admixed individuals are sampled at present times, and 50 time points are considered for all ancestral samples. Dystruct was run with $N=5000$ (running time: 50 minutes for each data set). FA running time: 0.11 second for each data set. Structure-snmf running time: 0.12 seconds for each run and each data set.

We modified the manuscript as follows:

updated citation:

Joseph, T. A., & Pe'er, I. (2019). Inference of Population Structure from Time-Series Genotype Data. *The American Journal of Human Genetics*, 105(2), 317-333.

added citation:

Harney, É., Patterson, N., Reich, D., Wakeley, J. (2020). Assessing the Performance of qpAdm: A Statistical Tool for Studying Population Admixture. bioRxiv. doi: 10.1101/2020.04.09.032664

Q3. Figure 1A) I am having some problems interpreting this plot. In the label of the Figure the authors mention: "Samples with ages ranging from 0 (present-day, grey color) to 4,000 generations (past, dark blue color) were simulated. A) Covariance matrix for observed samples," If I am interpreting Figure 1A) correctly, the colors for each square shown in the covariance matrix are not related to the ages of the sample, but to the covariance between a pair of samples. What is the interpretation that should be given to each color? Is the color related to the amount of covariance between a pair of samples? Also, should I interpret that there is a low covariance between samples taken at 4 ky and all of the samples because there is a predominant blue color in the comparisons between samples taken at all ages and the sample taken at 4ky?

Answer to: *What is the interpretation that should be given to each color?*

In the PCA and FA plots (Figure 1B-1D) the colours were set to represent the sampling ages. In the covariance plots (Figure 1A-1C) the colours represented covariances between pairs of samples. To clarify this point, we modified the caption of Figure 1 as follows

"A-C) Coalescent simulation of allele frequencies drifting through time in a random mating population. Samples with ages ranging from 0 to 4,000 generations were simulated. A) Covariance matrix for observed samples. In covariance matrices (A and C), the blue color indicates lower covariance values whereas the yellow and grey colors indicate higher values. B) FA and PC plots of samples (present-day: grey color, past: dark blue color), C) Brownian covariance matrix defined as the minimum of transformed times for pairs of samples, D) FA and PC plots for a coalescent simulation of a two-population model (left: population 1, right: population 2, present-day: grey color, past: dark blue color). E-F) Factor analysis corrects for a shrinkage effect visible in projections of ancient samples (salmon and blue colors) onto PCs of present-day admixed individuals (yellow color). Grey crosses represent population centers from which admixture estimates are computed."

Answer to: *Is the color related to the amount of covariance between a pair of samples?*

Exactly. The connection between the sample covariance in Figure 1A and the 'theoretical' model in Figure 1C can be motivated by the following (classical) arguments. Suppose we have two individuals sampled at distinct times, s and t . For individual 1, an observed genotype consists of p centered allele frequencies $x = x_1, \dots, x_p$, across the genome. For individual 2, the genotype is $y = y_1, \dots, y_p$. The colour gradient measures the covariance between the two genotypes, $\text{cov}(x,y) = 1/p \sum x_j y_j$.

We assume a Brownian model for each allele frequency, $x_i \sim B_s$, and $y_j \sim B_t$, where times are linearly transformed from sample ages and rescaled in the (0,1) interval and 1 is present-day. According to the law of large numbers, we have

$$\text{cov}(x,y) = 1/p \sum x_j y_j = E[B_s B_t] = \min(s,t) \quad (\text{eq *}).$$

Equation (*) establishes the connection between Figure 1A and 1C mathematically.

Answer to: *Also, should I interpret that there is a low covariance between samples taken at 4 ky and all of the samples because there is a predominant blue color in the comparisons between samples taken at all ages and the sample taken at 4ky?*

Exactly. According to eq (*), this corresponds to an expected result. The age 4ky corresponds to the oldest samples and to smallest transformed time, s , measured on a forward time-scale. There is a blue colour, because s is the minimum in all pairwise comparisons.

We modified the following sentence in the text, **Line 133 Page 7**

“The patterns observed in the sample covariance matrix were highly similar to those obtained in a theoretical covariance model corresponding to a Brownian process (Figure 1AC). In this model, pairs of samples that included the most ancient sample had the smallest covariance value.”

Q4. Page 7 Line 126. “The patterns observed in the sample covariance matrix were highly similar to those obtained in a theoretical covariance model corresponding to a Brownian process” Please expand more on the methods on how you obtained the theoretical covariance model corresponding to a Brownian process.

Although modelling allele frequencies with a Brownian process is at the root of recent population genetic methods, it is a very old idea. Cavalli-Sforza and Edwards (1967) wrote that “the distribution of transformed allele frequencies are approximately Gaussian with variance proportional to the time elapsed, while the mean will be constant in the absence of directional selection. We have already decided not to consider individual directional selection of a prolonged nature [...]. Thus, the random walk of the gene frequencies may be regarded as a Brownian-motion or Wiener process in the transformed space. The simplest model is one in which the Brownian motion has a constant rate which is the same for all characters at all times. This represents the case in which random drift alone determines the variation in gene frequencies, and population size and structure are taken to be constant.”

The covariance function for the Brownian process is $\text{Cov}(B_s, B_t) = \min(s, t)$, see eq (*) in our answer above. It is a classic result that can be found in many textbooks on stochastic processes. We added a reference to a readable textbook on stochastic processes **Line 378 Page 34**.

We modified the manuscript **Line 97 Page 6**

“The Brownian motion may be the simplest model in which the allele frequencies are approximately Gaussian with variance proportional to the time elapsed, while the mean remains constant in the absence of selection. In this case, random drift alone determines the variation in gene frequencies, and population size and structure are taken to be constant (Cavalli-Sforza and Edwards 1967)”

Added references:

Cavalli-Sforza, L. L., Edwards, A. W. (1967). Phylogenetic analysis: models and estimation procedures. *Am J Hum Genet*, 19, 233-257.

Ross, S. M. *Stochastic processes*. New York: Wiley, 1996

Q5. Page 10 Line 167. “we performed simulations of ancestral and admixed genotypes based on latent factor models.” Please provide more details on methods on how this is done.

We modified the manuscript **Page 10 Line 180**.

“Based on connections between population genetic models and matrix factorization [cite{Patterson2006, McVean2009, Engelhardt2010}], we performed simulations of ancestral and admixed genotypes based on factor models. The generative mechanism was similar to the simulation of multilocus genotypes in STRUCTURE, based on a Q-matrix of individual mixture coefficients

multiplied by a matrix of allelic frequencies in source populations \cite{Pritchard2000, Engelhardt2010}. By replacing allelic frequencies with Gaussian variables and using a probit link function, our method had the advantage of providing a ground truth both for factors and for admixture coefficients, while conserving interpretability in terms of standard population genetic concepts.”

Added reference

Engelhardt, B. E., Stephens, M. (2010). Analysis of population structure: a unifying framework and novel methods based on sparse factor analysis. *PLoS Genetics*, 6(9).

We modified the method section **Line 441 Page 27** as follows

“We generated genotypes for ancestral and admixed individuals by using factor models with a probit link function. Factor models are generative models close to PCA and STRUCTURE \cite{Engelhardt2010, Pritchard2000}. While matrix factorization underlies inference methods in STRUCTURE and in PCA, these models can also be used to generate new data. Considering two ancestral populations, the relative distance of population centers on the first factor axis in the simulation was proportional to the square root of the divergence time \cite{McVean2009}. Samples from the admixed population were generated at distance between the ancestral population centers proportional to the ancestry shared with each source. The generative model incorporated a time frame for simulating samples at distinct time points based on equation (1). In those simulations, the timing of the admixture event and the effective population sizes were reflected into the drift coefficient, λ (or equivalently, α).”

Q6. Figure 1D) The authors mention that “In a PCA of simulated samples, PC1 reflected the level of divergence between populations” Am I correct in interpreting that the samples with a PC1 value smaller than 0 belong to one particular population while the samples with a PC1 value bigger than 0 belong to the other population? If that is the case, it would be helpful to label the samples with a different shape. As an example, the samples in the left side of the plot could be labeled with a star while the samples in the right side could continue to be labeled with the circle.

It is correct to interpret samples with a PC1 value smaller than 0 belonging to one particular population while the samples with a PC1 value bigger than 0 belong to the other population. This interpretation is supported by coalescent theory (McVean, 2009) and by statistical theory (Ding and He 2004).

We modified **Figure 1D** to clearly indicate which samples belong to which population. Since there are only two populations, we chose to label them pop 1 & pop 2 in the figure. We included a citation of McVean’s work in the text, but found (Ding and He 2004) perhaps too abstract for our presentation (and we did not include this reference).

Ding, C., He, X. (2004). K-means clustering via principal component analysis. Proceedings of the twenty-first international conference on Machine learning.

Q7. Page 40 Line 672 “Prior to PC and FA analysis, the matrix of genotypes was corrected for the effect of genomic coverage by fitting a latent factor mixed model with coverage considered as the primary variable (Caye et al. 2019).” Please provide more details on how this is done.

We modified the manuscript **Line 557 Page 32** (Methods).

“We adjusted the centered matrix of genotypes for coverage by performing a regression analysis of the form $Y_{ij} = a_j \text{coverage}_i + \tilde{Y}_{ij} + e_{ij}$. The regression analysis assumed that

lower coverages could correlate with particular population samples. The correction was performed by fitting a latent factor mixed model with coverage considered as primary variable \cite{Caye2019}. In subsequent analyses, the genotype matrix was replaced by the latent matrix \tilde{Y}_{ij} adjusted in the regression model."

Note: The samples were genotyped by distinct research groups and with various genotyping efforts. We used LFMM 2 to remove potential batch effects. We have included a function of our program that allows users to perform correction for coverage without calling external packages: https://bcm-uga.github.io/tfa/reference/coverage_adjust.html

We believe that this apparently minor point is a great advantage for PCA and factor models over methods like STRUCTURE. PCA and factor models can handle matrices other than allelic or genotypic frequencies. In particular, data matrices adjusted for various experimental biases can be analysed with PCA and FA, while it is much more difficult to do it with STRUCTURE or qpAdm.

Q8. Provide a little bit more of information on the README.md file from <https://github.com/bcm-uga/tfa> on how to run the methods. I recommend to list what methods are contained in the repository and giving a small manual on how to run them.

We modified the README file in GitHub as suggested. The package has a web site, and a link is now provided in the GitHub source repository <https://bcm-uga.github.io/tfa/index.html>. We also plan to release it on CRAN.

We wrote a tutorial showing how the main functions can be run on an example related to the manuscript's contents. Note that GitHub doesn't allow their users to deposit large data sets. For this reason, the sample size of the example, simulating admixture results for UK Bell Beaker samples are much lower than in the paper.

Q9. Figure 1A) The Y-axis label should indicate the sample age in ky.

Page 4 Line 64.- "In this study, we provide evidence that representations of ancient DNA samples based on PC projections, STRUCTURE or PCA disagree with each other and with F-statistics. We introduce a factor analysis that enables geometric representations of ancient samples in better agreement with the later" Does 'later' refer to F-statistics here?

We modified the x-axis in Figure 1A and the axes in Figure 1C. In Figure 1A, the rows and columns represent samples. Each sample had a unique sampling date, and we identified the samples with their dates. In Figure 1C, the rows and columns represent ages in the theoretical model. Thus, it was more accurate to label the axes as 'sample ages'.

We modified the text **Page 4 Line 64**.

"We introduce a factor analysis that enables geometric representations of ancient samples in better agreement with F-statistics."

Q10. Page 12 Line 185 "Considering a lower level of divergence between ancestral populations did not change the ranking of method performances (Figure S3)." What is the level of divergence on Figure S3 compared to Figure 2?

The low level of divergence was an $F_{ST} = 5\%$ while the high level corresponded to an $F_{ST} = 25\%$. We modified the text **Page 12 Line 185**.

“Considering a lower level of divergence between ancestral populations did not change the ranking of method performances (Figure S3, $F_{ST} = 5\%$)”.

In Caption of Figure 2: “models with high level of divergence between the ancestral populations ($F_{ST} = 25\%$)”.

Q11. Page 12 Line 187 “When the drift parameter decreased, estimates from PC projections exhibited less bias, and became closer to qpAdm values (Figure S3B).” It is hard to observe this from Figure S3B. If the authors wish to make this point clearer, it would be good to make an additional boxplot (or a similar plot) that proves this claim.

We reported the values of the biases mentioned and added a significance value in the text. We modified the manuscript **Page 12 Line 200**

“For weak drift, estimates from PC projections exhibited less bias than qpAdm values, while keeping similar levels of statistical error (Figure S3B, bias for PCA: 2.3%, bias for qpAdm: 4.9%, $t = 4.02$, $P = 1e-4$).”

Q12. Page 12. Line 190 “We eventually performed “realistic” simulations with low differentiation, low drift, and relatively recent mixture consistent with the demography of ancient Europeans (Figure S4)”. I recommend providing a figure in S4, or at least an explanation in the caption, of the scenario you are simulating that is consistent with the demography of ancient Europeans.

The simulated scenarios were consistent with summary statistics obtained when studying two-way admixture models of European Bronze Age populations:

- $F_{ST}(\text{Yamnaya, Anatolia}) = 0.049$ (uncorrected estimate) estimated from 16 Pontic steppe and 19 Anatolian samples, and it was equal to 0.05 in the simulations.
- Time of admixture = 5ky BP corresponded to $(5000 - 3964)/(8300 - 3964) = 23.9\%$ of the earliest sample relative to the most recent one, and this proportion was set to 25% in the simulations
- The estimated noise-to-drift ratio (drift parameter), λ , was around 0.1 in German Bell Beakers, 0.05 in Czech Bell Beakers, and 0.01 in England Bell Beakers. Because higher values of λ indicate weaker drift values, we set $\lambda = 0.15$ in simulations as a conservative choice when comparing methods accounting for drift over the other methods.

We modified the caption of Figure S4 as follows:

“Comparisons of ancestry estimates for factor analysis (FA), qpAdm, projections on PCs of admixed samples, and STRUCTURE implemented with sparse NMF. The level of divergence between ancestral populations was equal to $F_{ST} = 0.05$, the time since admixture was around 0.25 of the oldest sample date, and the amount of drift was low ($\lambda = 0.15$). These values were related to the admixture of European Bronze Age populations, considering a date of admixture around 5ky BP, an uncorrected F_{ST} between Yamnaya and Anatolian samples as observed in the data, and an upper bound on noise-to-drift ratios.”

Q13. Page 17 Line 252 “A two-source model of ancestry was assumed in each region, and the drift parameter was chosen to remove temporal effects from factor 2 (Figure S9).” Please expand on how this is done on the caption of Figure S9.

We modified the caption of Figure S9 as follows:

“Factor analysis of samples from four geographic regions, considering Anatolian Neolithic and Yamnaya steppe samples as sources in two-way admixture analyses. In those analyses, regression analyses were performed with sample age considered as a response variable explained by latent factor 1 and 2. The drift parameter (λ) was selected as the largest value for which the variance explained by factor 2 was removed.”

We also modified the computer program, documentation files and tutorial to add a new function that performs the regression analyses shown in Figure S9.

See a tutorial: <https://bcm-uga.github.io/tfa/articles/tfa-vignette.html>
and function documentation: https://bcm-uga.github.io/tfa/reference/choose_lambda.html

Q14. Page 30 Line 464.- “The dates are normalized to fall the unit interval $0 < t_1 \leq \dots \leq t_n \leq 1$.” I believe the inequality should be equal to $0 \leq t_1$.

We modified the text **Page 30 Line 464**.

We changed to $0 \leq t_1$, however the equality case would happen only in the case where a sample has the exact same centered allele frequency at all SNPs, leading to a variance of zero.

Q15. Page 35 Line 561.- “we generated genotypes for ancestral and admixed individuals from a factor model with a probit link function, providing simulations for which a ground truth was available for the comparisons of methods.” How does this work? Please provide more details.

We detailed the generative simulation procedure as well as the updated sections (methods and main text) in our reply to Q5.

We appreciate your efforts spent in evaluating our manuscript, especially in a period during which many reviewers had to decline reviews. We found the comments constructive and very useful.

Reviewer #2:

François and Jay present a very useful new method for controlling for the effects of temporal drift when performing exploratory data analysis of population structure on ancient genomes. I think the methodology is clearly laid out, the figures are beautiful and I appreciate that the code is made readily available for potential users. I would recommend this paper for acceptance with only a few minor edits. I mostly have comments and suggestions for text clarifications. I’m happy if the authors think certain analyses might require lots of work and should be left to a future study.

We thank you for positive comments and for recognition of the efforts we made to provide codes for potential users. Here is a summary of the changes made to the manuscript to account for your suggestions.

We answered the question about the robustness of the method to sample age uncertainty and sample sizes (Q1-Q2), and we included a supplementary Table that evaluates the robustness of ancestry estimates to uncertainty in estimated ages (Table S3, see below).

We clarified the text at several places (Q3-Q5). We explained why we considered that visual representations of ancient samples should agree with F-statistics. Our claim is also supported by a recent preprint by Harney et al. (2020). We also explained the similarities and differences with the LFMM approach presented in (Caye et al. 2019), and provided additional comments on Figure 1.

To answer Q6, we ran additional analyses to evaluate the proportion of variance explained by the Brownian model in two-way admixture analyses of European samples (Neolithic, Bronze Age and more recent samples). We extended the package “tfa” to implement those computations, and included a supplementary Table for the proportions of variance explained by the Brownian covariance model in two-way admixture estimates (Table S2, see below).

Regarding Q7-Q8, we feel that using factor analysis (or PCA) to estimate population size variation over time or to detect selection acting on genetic variants goes beyond the scope of our study and would require too many additional analyses for a single paper. There are also many difficulties in doing this, and we think that these questions should be addressed in another study.

Q1. How does your method perform if the ages are misspecified? In other words, how robust is it to mis-dated samples at different levels of mis-dating? How did you deal with ranges of C14 dates? Did you take their midpoint?

In our analyses, sample ages were taken from the published studies introducing the corresponding ancient samples as “average of 95.4% date range in calBP (defined as 1950 CE)”. These dates were available from a metadata file downloaded from D. Reich lab’s website. In our study we used the point estimates provided in the metadata file.

We added a supplementary Table and modified the text **Line 583 Page 33**

“Regarding sample ages, we used the central estimates provided in original studies. To check the robustness of our analyses to uncertainty in sample ages, we replicated analyses using random ages drawn from reported confidence intervals. More specifically, we created 100 replicates for each data set in two-way admixture analyses of European populations. The confidence intervals spanned around 400 years, and the ages were randomly drawn around the central estimate with a standard deviation of 100 years. Resampling individual ages from their confidence intervals biased the admixture coefficients toward intermediate values (50% Yamnaya - 50% Anatolia). The bias was around 1%, and it remained moderate for all studied populations. The projections of individuals on factor 1 were highly stable (S.D. over 100 replicates ranged between 0 and 1.3%), and the squared correlation of the replicated factor 1 with the factor obtained using the central estimate was around 99.9%. These results showed that for ancient European samples, FA results were robust to moderate levels of uncertainty in sample date estimates (Table S3).”

Now included in Supp. Mat:

Table S3: Estimates of Yamnaya ancestry in two-way admixture models for European populations. FA scores and admixture coefficients using randomized dates (SD = 100y, 100 replicates) are compared to max-likelihood age estimates.

	admixture std dev.	bias toward 50%	Sq. score correlation
Germany_LBK_N	0.0%	-0.1%	99.9%
Austria_LBK_N	0.0%	0.0%	99.9%

Greece_Peloponnese	0.0%	0.0%	99.9%
England_EBA	1.3%	1.8%	99.9%
Czech_R_EBA	0.9%	1.3%	99.9%
Germany_EBA	0.5%	0.7%	99.9%
Germany_CW	0.1%	0.1%	99.9%
England_BA	1.5%	1.5%	99.9%
Czech_R_BA	1.2%	1.8%	99.9%
Germany_BA	0.9%	2.6%	99.8%
Netherlands_BA	1.3%	1.5%	99.9%
Sweden_Vikings	0.2%	0.7%	99.9%
England_Saxon	0.2%	0.0%	99.9%
Hungary_Langobard	0.2%	0.5%	99.9%
Italy_Langobard	0.1%	-0.2%	99.9%

admixture std dev: std deviation of population average admixture coefficients over 100 runs with uncertain ages.

bias toward 50%: observed bias compared to the estimate obtained as average of 95.4% date range in calBP (defined as 1950 CE), a positive value means that the estimate was biased toward 50% and a negative value means bias toward 0 or 100%.

Sq. score correlation: Average of squared correlation between factor 1 scores w/o age uncertainty.

Q2. Is the utility of the method dependent on sample size? You apply this to a dataset of hundreds of human genomes. If you were to apply it to a dataset of a few genomes (e.g. by subsampling your current data), would you perform as well as PCA, better or worse?

While our Figure 3 presented results for 704 ancient Eurasian individuals with ages ranging between 0.4 and 14ky BP, the rest of the manuscript (Figures 4-6) detailed analyses realised with smaller sample sizes, often less than 100 individuals including target and source samples. In all those analyses, we subsampled the current data, and compared the results of FA with PCA. Our study showed that FA results were generally better than PCA projections (Figures S15 and S16). PC projections are usually fine when present-day admixed samples are projected onto axes built from source populations, but their current use with ancient samples is at the opposite of this practice (ancient samples are projected onto axes built from present-day admixed samples). Thus PCA projections could be (very) sensitive to the choice of reference populations (Figure 1).

In addition, PCA, or at least low-rank PCA, could be viewed as a limiting case of our FA method for large values of lambda, ie, for very small drift or for contemporary samples. Since PCA is a particular case of FA, FA could theoretically not do worse when the drift parameter (lambda) is correctly evaluated. We have modified our computer program to provide an additional function that evaluates the choice of the drift parameter in the FA model by using the simple heuristic described in our M&M section.

We modified the manuscript **Line 406 Page 25**

“In principle, FA recasts the PCA problem as an estimation problem in an unsupervised regression model. In terms of sample sizes, the limitations of FA are similar to those of linear regression, ANOVA or PCA”.

Q3. Could you motivate a bit more why agreement with F-statistics / qpAdm should be seen as the “gold standard”?

We acknowledge that the expression “gold standard” may not be the most appropriate one. By “gold standard”, we meant an accurate “state-of-art” method based on a model of genetic drift. We think that the recent preprint by Harney et al (2020) provides additional support that ancestry estimates obtained with qpAdm are generally correct. The main difference with our approach is that qpAdm is only statistical and does not attempt to represent individual samples in a geometric space.

The program qpAdm computes a matrix of f4-statistics in which target and source populations are introduced as ‘left’ populations. Additionally, users must specify a set of ‘right’ populations that serve as references or outgroups against which the relationships of the target and source populations are considered. F4-statistics rely on shared genetic drift between sets of populations and a simplified model of population tree. The principle of F-statistics is close to the Brownian model hypothesis in which the population tree branch lengths are used to compute the variance-covariance model of allele frequencies. The performances of qpAdm are now better explained in a recent preprint by Harney et al. (2020).

We modified our main text **L178 Page 10**

“Because estimates accounted for shared genetic drift in `{\tt qpAdm}`, this approach was considered to be the “state-of-the-art” method for estimating admixture coefficients from ancient DNA samples `{\cite{Harney2020}}`.”

Added reference

Harney, É., Patterson, N., Reich, D., Wakeley, J. (2020). Assessing the Performance of qpAdm: A Statistical Tool for Studying Population Admixture. bioRxiv. doi: 10.1101/2020.04.09.032664

Q4. Can you elaborate on similarities / differences between this model and the LFMM model for environmental association? It seems like this is similar to the models in Frichot et al. and Caye et al.?

LFMMs are latent factor regression models in which a (centered) data matrix is explained by some explanatory variables (Z) and unobserved factors (U). The effect sizes of explanatory variables, latent factors and their loadings are estimated simultaneously by solving least-squares estimation problems. Frichot et al. (2013) and Caye et al. (2019) used LFMMs to estimate the effect sizes for the environmental variables, thus performing a particular type of genome-wide association studies. They considered the latent factors as nuisance variables which are only useful in corrections for confounding.

The FA model has no explanatory variables, but instead considers a covariance model for the random part of genetic variation. Put in the Bayesian framework, the estimation problem can be reformulated as an LFMM, in which Z are defined as the principal components of the covariance model. The respective roles of Z and U are interchanged. The latent factors are no longer considered to be the confounders, but they become output of the analysis. The algorithm is not exactly the same as in Caye et al. (but it is very close). In particular the factors returned by the `lfmm()` function in R are

not orthogonal factors (this is not needed for GWAS applications), and FA need to perform an additional factorisation step compared to LFMM.

Q5. In Figures 1A and 1C, what do you mean in the y-axis label when you say “Samples”? As this is a covariance matrix, I’m guessing this should be the same label as the x-axis, and should say “Sample age (ky)” as well? Can you clarify in the caption to Figure 1C which matrix this is in your model? I’m guessing matrix C but it doesn’t seem to represent $\min(t_i, t_j)$ in each of its cells, so I’m a bit confused as to what this represents.

(Note: Reviewer’s Q5 and Q6 were copied in a reverse order for clarity)

Q5.1-2: In this particular case, samples and sample ages are exchangeable, because each sample has a unique age in the simulation. Still, we changed the x-axis label as “Samples” in 1A and the y-axis label as “Sample age (ky)” in 1C as suggested.

Q5.3: The matrix C in Figure 1C indeed represents $\min(t_i, t_j)$ where ages were rescaled/redefined as “forward” times (0= age of MRCA, 1 = present-day)

We modified **Figure 1A and 1C** and the **caption of Fig. 1** to clarify what was actually computed in Fig 1C.

Q6. Is there information in the matrix B about SNPs that vary strongly with time, and could potentially be used to detect on individual variants over time?

Q7. In some sense, the temporal drift you’re correcting away is also an important feature of the data. Could estimates of temporal drift be represented in separate plots when studying real data, along with the factor plots?

The FA model corrects for distortions in PCA due to drift, but does not remove all effects of drift. For example, population divergence caused by genetic drift is reflected in the loading matrix V computed in the FA. The B matrix mainly contains uninterpretable coefficients, that we considered as nuisance variables in analyses. These coefficients can however be used to evaluate the overall proportion of variance explained by distortions in a standard PCA.

We extended the R package tfa by adding a function that returns a vector of squared correlation coefficients representing the proportion of variation captured by the specified covariance model at each locus. We used it to compute the proportion of variance explained by the covariance model for ancient European populations in two-way admixture models, and found that the mean values averaged over genetic variants and individuals were around 20% (Table below).

The histograms showing the distribution of variance across SNPs suggested departure from the empirical null beta distribution with mean < standard error expected under neutrality. Heavy tails in the distribution of R^2 (mean = standard error) could be explained by natural selection acting on loci to change the “effective population size” at these loci. The function could be used to detect variants that differ from background genomic variation, perhaps due to a lower effective population size or a signature of positive selection. We did not develop this point further in the manuscript because of the lack of space and difficulties to discuss this point in relation with ancestry and geometric representations of samples. Nevertheless, we feel that Table S2 provides additional evidence that our corrections removed a substantial fraction of undesired signals in the data.

We modified the manuscript **Line 292 Page 19**

“In two-way admixture analyses, the estimates of FA and qpAdm strongly agreed with each other (Figure S10, Figure S11), providing evidence that positions on factor 1 corresponded with steppe ancestry. In FA, the proportion of variance explained by temporal distortions was non-negligible (mean: 17.7%, SD: 7.4%, Table S2).”

Table included in Supp. Mat:

Table S2: Proportion of variance explained by the first eigenvectors of the covariance model in two-way admixture estimates (Anatolia and Pontic steppe as source populations).

	Proportion of variance explained by distortions	Standard error
Germany_LBK_N	10.6%	10%
Austria_LBK_N	9.5%	9.9%
Greece_Peloponnese_N	39.9%	22.6%
England_EBA	20.9%	19.6%
Czech_R_EBA	17.1%	16.7%
Germany_EBA	13.4%	14%
Germany_CW	7.6%	8.5%
England_BA	20%	19.6%
Czech_R_BA	22.1%	20.7%
Germany_BA	15.6%	15.7%
Netherlands_BA	18.2%	17.9%
Sweden_Vikings	17.2%	17.2%
England_Saxon	19%	17.3%
Hungary_Langobard	17.7%	17.3%
Italy_Langobard	17.4%	17.6%

Q8. Could the estimates of temporal drift be used as a measure of population size changes over time? (essentially looking at how the temporal variance in allele frequencies for a SNP relate to the observed dates)

As discussed in our answer to Q7, our model does not provide any direct estimate of drift or effective population size. Doing this would be difficult with FA, and we need (a lot of) time to answer this question that, in our opinion, goes beyond the scope of our study.

Q9. Line 39, page 3: "counterpart" should be plural

This grammatical error was corrected.

Signed: Fernando Racimo

We appreciate your time and efforts spent in evaluating our manuscript, especially in a period during which many reviewers had to decline or delay their reviews. The comments were constructive and very useful. We are also honored that your signature appeared with the review.

REVIEWERS' COMMENTS:

Reviewer #1 (Remarks to the Author):

The authors have answered all my questions. I thank the authors for the very clear and detailed response to my comments. The method presented by the authors is an important contribution to the analysis of past demographic events using ancient DNA. I strongly recommend the publication of this paper.

Diego Ortega-Del Vecchyo

Reviewer #2 (Remarks to the Author):

The authors have addressed all of my concerns. I recommend this paper for acceptance.